# Degrading permafrost puts Arctic infrastructure at risk by mid-century

Jan Hjort [1], Olli Karjalainen [1], Juha Aalto [2,3], Sebastian Westermann[4], Vladimir E. Romanovsky[5,6], Frederick E. Nelson[7,8], Bernd Etzelmüller[4] & Miska Luoto [2]

Degradation of near-surface permafrost can pose a serious threat to the utilization of natural resources, and to the sustainable development of Arctic communities. Here we identify at unprecedentedly high spatial resolution infrastructure hazard areas in the Northern Hemisphere's permafrost regions under projected climatic changes and quantify fundamental engineering structures at risk by 2050. We show that nearly four million people and 70% of current infrastructure in the permafrost domain are in areas with high potential for thaw of near-surface permafrost. Our results demonstrate that one-third of pan-Arctic infrastructure and 45% of the hydrocarbon extraction fields in the Russian Arctic are in regions where thaw-related ground instability can cause severe damage to the built environment. Alarmingly, these figures are not reduced substantially even if the climate change targets of the Paris Agreement are reached.

[1] Geography Research Unit, University of Oulu, P.O. Box 3000, Oulu 90014, Finland. [2] Department of Geosciences and Geography, University of Helsinki, P. O. Box 64, Helsinki 00014, Finland. [3] Finnish Meteorological Institute, Weather and Climate Change Impact Research, P.O. Box 503, Helsinki 00101, Finland. [4] Department of Geosciences, University of Oslo, P.O. Box 1047, Oslo 0316, Norway. [5] Geophysical Institute, University of Alaska Fairbanks, Fairbanks AK 99775 AK, USA. [6] Earth Cryosphere Institute, Tyumen Science Centre, Siberian Branch of the Russian Academy of Science, Tyumen 625026, Russian Federation. [7] Department of Geography, Environment, and Spatial Sciences, Michigan State University, East Lansing MI 48824 MI, USA. [8] Department of Earth, Environmental, and Geographical Sciences, Northern Michigan University, Marquette MI 49855 MI, USA. Correspondence and requests for materials should be addressed to J.H. (email: jan.hjort@oulu.fi)

Arctic natural and anthropogenic systems are undergoing unprecedented changes[1], with permafrost thaw as one of the most striking impacts in the terrestrial cryosphere[2–4]. In addition to the potential adverse effects on global climate[5], ecosystems[6], and human health[7], warming and thaw of near-surface permafrost may impair critical infrastructure[8,9]. This could pose a serious threat to the utilization of natural resources[10], and to the sustainable development of Arctic communities[9,11,12]. Extensive summaries of damage to infrastructure along with adaptation and mitigation strategies are available[11,13–18]. Benchmark reports[1,13,14] call for pan-Arctic geohazard explorations and infrastructure risk assessments, but only regional studies[17,19–21] have been conducted since ref. [8]. There is an urgent need for pan-Arctic geohazard mapping at high spatial resolution and an assessment of how changes in circumpolar permafrost conditions could affect infrastructure[1,14]. Owing to the increasing economic and environmental relevance of the Arctic[1,5,10], it is of a vital importance to gain detailed knowledge about risk exposure in areas of current and future infrastructure[8–14,18]. The aim of this study was to first, map infrastructure hazard areas in the Northern Hemisphere's permafrost regions at unprecedentedly high (~1 km) spatial resolution under projected climatic changes and second, quantify the amount and proportion of engineering structures in areas where ground subsidence and loss of structural bearing capacity could damage infrastructure by 2050.

Standardized in-situ observations of ground temperature and thaw depth, geospatial environmental data, and ensemble methodologies[22] were used to model the current and future ground thermal regime in the Northern Hemisphere's permafrost domain[23] (Methods). In the ensemble forecasting, the observations of ground thermal regime were related to physically relevant environmental variables using four statistical techniques at 30 arc-second resolution. The ensemble model provided accurate predictions of the observed ground temperatures retrospectively, indicating potential for good performance in prognostic studies[23]. The focus was on thaw of near-surface permafrost (<15 m depth) owing to its central role for infrastructure (defined here as facilities with permanent foundations on ice-free land) hazards[1,8,9,13,14]. Future conditions were considered using Representative Concentration Pathways[24] (RCPs) 2.6, 4.5 and 8.5 for 2041–2060 and 2061–2080. Near-surface permafrost was considered to thaw when mean annual ground temperature (MAGT) at or near (the closest to) the depth of zero annual amplitude changed from ≤0 °C to >0 °C between the baseline (2000–2014) and future period. In an engineering context, the selected threshold is conservative because infrastructure (e.g., buildings) could experience thaw settlements and failure before the thaw of near-surface permafrost. However, a conservative threshold is justified considering the use of statistically based methodology in modelling of the ground thermal regime (Methods). In the results, we focus on medium stabilization scenario RCP4.5 (total radiative forcing is stabilized shortly after 2100)[25] for 2041–2060 (hereafter 2050). This period was chosen acknowledging the relatively short lifespan (often 20–50 years) of infrastructure in the Arctic[12,13]. Thus our analyses, conducted at unprecedentedly high spatial resolution, are useful for indicating near-future at-risk areas owing to the degradation (i.e. warming and thaw) of permafrost.

Using the forecasts of the ground thermal regime[23], a consensus of three geohazard indices (Methods), and infrastructure data products we identified central infrastructure hazard areas (i.e., areas of near-surface permafrost thaw and high hazard indicated by the consensus index) and quantified infrastructure elements potentially at risk owing to climate change. We focused on current infrastructure fundamental to Arctic communities and economic activity, including residential (settlements and buildings), transportation (roads, railways and airports) and industrial facilities (pipelines and industrial areas)[9,13,14,26]. Moreover, we considered current (2015) pan-Arctic population and hydrocarbon extraction fields in the Russian Arctic as special investigation targets.

Our study reveals the magnitude of the threat to engineering structures from climate change at the pan-Arctic scale, and shows where detailed infrastructure risk assessment should be conducted in the near future. Our results demonstrate that ca. 70% of current infrastructure in the permafrost domain is in areas with high potential for thaw of near-surface permafrost by 2050. One-third of the pan-Arctic infrastructure and 45% of the hydrocarbon extraction fields in the Russian Arctic are located in high hazard regions where the ground is susceptible to thaw-related ground instability. The results show that most fundamental Arctic infrastructure will be at risk, even if the Paris Agreement target is achieved.

## Results

**Population and infrastructure in areas of permafrost thaw.** Results show that by 2050 3.6 million people, which constitutes about three quarters of the current population in the Northern Hemisphere permafrost area, may be affected by damage to infrastructure associated with permafrost thaw (Fig. 1, Supplementary Figs. 1 and 2, Supplementary Table 1). A substantial proportion of the fundamental human infrastructure is potentially under risk: 48–87% (mean = 69%) of the current pan-Arctic infrastructure is located in areas where near-surface permafrost is projected to thaw by mid-century (Fig. 2a, Supplementary Tables 2 and 3, Supplementary Data 1) (results for 2061–2080 are presented in Supplementary Fig. 3). The potential risk to railways appears to be especially high, as for example 470 km of the Qinghai–Tibet Railway[27] and 280 km of the world's northernmost railway, the Obskaya−Bovanenkovo railway (Fig. 3) may occur in the areas of thawing permafrost. The figures for residential infrastructure and pipelines are also high (Supplementary Tables 1 and 2). There are currently more than 1200 settlements (ca. 40 with population more than 5000) in the zone where permafrost thaw is likely. Our results also indicate that central oil and natural gas transportation routes may be at considerable risk: 1590 km of the Eastern Siberia–Pacific Ocean (ESPO) oil pipeline, 1260 km of major gas pipelines originating in the Yamal-Nenets region, and 550 km of the Trans-Alaska Pipeline System (TAPS) are in the area in which near-surface permafrost thaw may occur by 2050 (Supplementary Table 1).

**High hazard zone.** The regions associated with the highest hazard are in the thaw-unstable zone characterized by relatively high ground-ice content and thick deposits of frost-susceptible sediments, as well as increased potential for permafrost thaw (Supplementary Figs. 1, 4). By 2050, these high hazard environments will contain a population of nearly a million and 25–45% (mean = 33%) of existing pan-Arctic infrastructure (Figs. 2b, 3). This zone includes, for example, more than 36,000 buildings, 13,000 km of roads, and 100 airports (Supplementary Table 2). Moreover, 45% of the globally important oil and natural gas production fields in the Russian Arctic are located in areas with high hazard potential because of adverse ground conditions and thaw of near-surface permafrost by 2050.

**Geographical differences in infrastructure hazards.** Changes in ground thermal regime and bearing capacity could be a more serious problem in central Asian mountainous regions and Eurasia than for North American residential and industrial

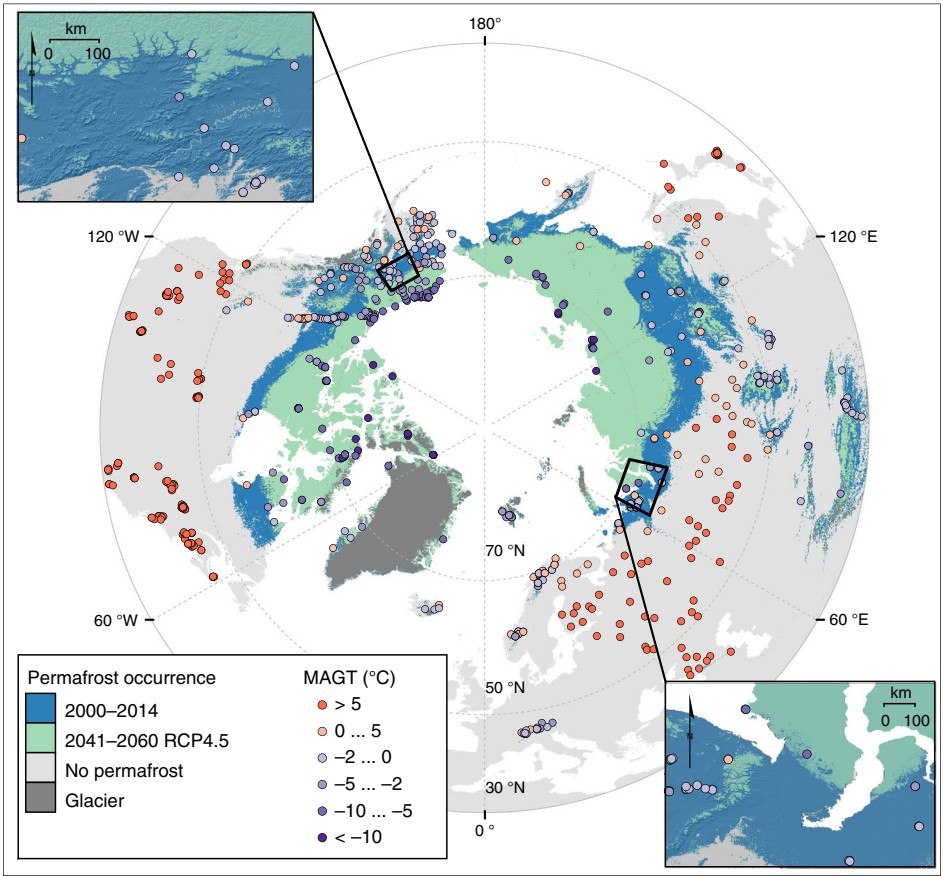

**Fig. 1** Distribution of permafrost in the baseline (2000–2014) and future (Representative Concentration Pathway 4.5 2041–2060) climates[23]. Note that the baseline extent of permafrost (blue) includes future distribution (greenish). The location and observed mean annual ground temperature (MAGT) of the data points (boreholes) are shown with coloured circles

infrastructure, if only permafrost thaw is considered (Fig. 2c). However, acknowledging a broader spectrum of factors known to contribute to the hazard level (Supplementary Fig. 1), the regional differences appear less pronounced (Fig. 2d). Still, most infrastructure types in Eurasia other than railways are more commonly threatened when compared to other regions. For example, our results highlight critical areas such as the Pechora region, the northwestern parts of the Ural Mountains, northwest and central Siberia, the Yakutsk basin (except for the city of Yakutsk in low to moderate hazard area), as well as the central and western parts of Alaska to which high priority for local scale infrastructure hazard assessments should be performed in future decades[1]. The Yamal-Nenets region in northwestern Siberia is important because it is the primary natural gas extraction area in Russia, and accounts for more than one-third of the European Union's pipeline imports[28] (Fig. 3).

**The effect of climate change scenario and model uncertainty.** Considering different stabilization scenarios of climate warming, our analysis reveals that substantial cuts in global greenhouse gas (GHG) emissions now would not make a large difference for infrastructure risks in the highest hazard area by 2050. This is because nearly the same number of buildings, roads, and other infrastructure would be jeopardized under moderate climate warming (RCP2.6) as compared to a pessimistic, business-as-usual scenario (RCP8.5) (Fig. 2b). This result is congruent with projected changes in the Arctic[1], and emphasizes the need for adaptation-based policies at community and regional levels in the near future. With respect to the decades following 2050, however,

attainment of the goals of the Paris Agreement's climate warming target (i.e., holding global warming to well below 2 °C above pre-industrial levels and pursuing efforts to limit the temperature increase to 1.5 °C)[29] would make a clear difference in terms of potential damage to infrastructure (Fig. 2b, Supplementary Fig. 3b).

Considering the uncertainty embedded in the projections of ground temperature and annual thaw depth, the quantity of infrastructure potentially under risk is probably not considerably smaller (lower limit), but could be substantially larger (upper limit) than estimated (Fig. 2b). For example, at least 19 large settlements (population > 5000) are predicted to occur in the highest hazard zone (estimate = 22), but the number could be as large as 34. This pattern results primarily from the overall reduction of human activity in the north, where the permafrost thaw potential is lower than in the warmer, southern regions[3,4]. The potential harm to industrial facilities could be larger than estimated (pipelines: estimate = 32%, range = 24–70%; industrial areas: estimate = 25%, range = 20–64%) (Fig. 2b). This is important because damage to pipelines and industrial facilities (e.g., stores of harmful substances) may lead to regional ecosystem disruption of major significance, such as large-scale oil spills[30]. Moreover, damage to critical energy delivery and industrial infrastructure can affect general economic activity and national security[11,13,14].

## Discussion
Degradation of permafrost has already been related to damage to thousands of infrastructure components[11,13,16,17] and negative

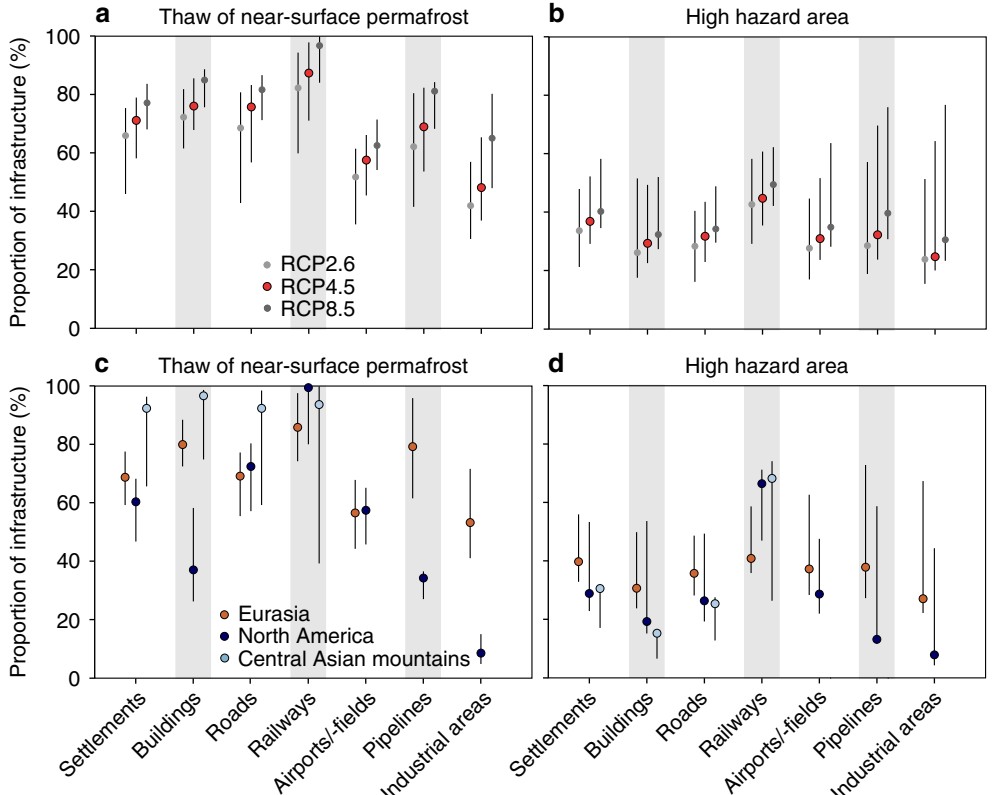

**Fig. 2** Central results of the infrastructure hazard computations. Proportion of all residential, transportation, and industrial infrastructure in areas of near-surface permafrost thaw (**a**) and high hazard (**b**) in the pan-Arctic permafrost area based on different Representative Concentration Pathway (RCP) scenarios by the middle of the century (2041–2060). The comparable results of RCP4.5 for geographical sub-regions (Eurasia, North America and central Asian mountains) are presented in (**c**) and (**d**) (also 2041–2060; percentages for airports/-field, pipelines and industrial areas are not shown for central Asian mountains owing to too few observations). The numerical results are presented in Supplementary Table 3. The uncertainty ranges (bars) were based on the uncertainty in the mean annual ground temperature (**a**–**d**) and active layer thickness predictions (**b** and **d**)

ecosystem impacts[1,13,30] across the Arctic. Detrimental effects on engineered structures, socio-economic activities, and natural systems can, therefore, all be expected throughout the permafrost domain under climate warming[1], particularly in high-risk areas with substantial urban and industrial centres such as Vorkuta and Novyi Urengoy in hot spots of infrastructure hazard in the Russian Arctic (Fig. 3). In addition to natural impacts, damage to infrastructure can be caused by anthropogenic factors, such as human-induced disturbance of the ground thermal regime and poor maintenance[13,14,31]. Although engineering solutions (e.g. adaptation strategies and structures such as insulation and thermosyphons that were not considered in this pan-Arctic study) can to some extent address both human-induced and naturally caused problems, their economic cost may be prohibitive at regional scales[9,14,32]. Consequently, detailed hazard maps and geospatial data-based computations, such as those presented here, are of importance to enable planners and policymakers to identify both high- and low hazard areas when planning future infrastructure at urban and settlement scales[2,9,17,33]. Our analyses were conducted at a higher spatial resolution than previous studies[17,19–21], and the results presented here are based on a consensus of three different indices (see Methods). Moreover, we were able to quantify and show the magnitude of infrastructure at risk across the circumpolar permafrost domain. The major advantage of the approach presented here is that hazard quantification can be conducted with any available infrastructure or population dataset (also using planned infrastructure and future population if suitable high-quality datasets and projections

are available) and for any policy-relevant global warming scenario.

Our study focused on a pan-Arctic assessment with the goal of showing where regional and local scale risk assessments, taking into account site-specific engineering, design, and construction practices (e.g., adaptation strategies)[2,9,11,13,14] should be conducted in the near future. The forthcoming infrastructure risk assessments would significantly benefit from applicable process-based transient models of ground thermal regime and high-resolution climate and ground-ice data. With the help of improved permafrost projections, hazard maps and verified infrastructure data, it would be feasible to quantify the economic impacts of climate change on infrastructure at the pan-Arctic scale (e.g., following ref. [9]).

To successfully manage climate change impacts in sensitive permafrost environments, a better understanding is needed about which elements of the infrastructure are likely to be affected by climate change, where they are located, and how to implement adaptive management in the most effective way, considering the changing environmental conditions. Such locally and regionally applied mitigation strategies for existing infrastructure and future development projects are paramount for sustainable development in the Arctic[1]. Our study can be considered to be a step forward toward these goals.

In conclusion, this is the first study to explicitly show the amount of fundamental infrastructure potentially at risk across the Northern Hemisphere permafrost area under climate change. A total of 69% of the pan-Arctic residential, transportation, and

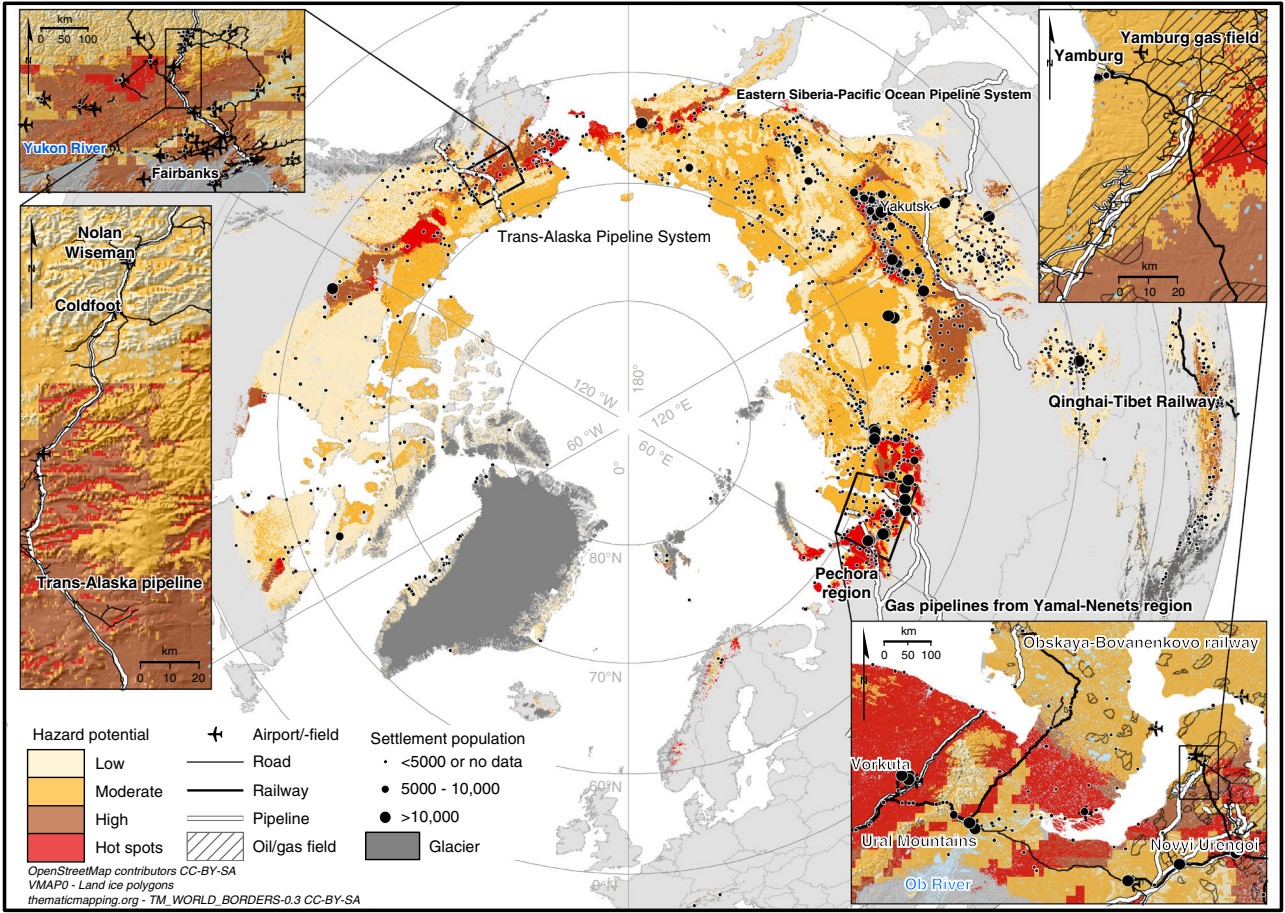

**Fig. 3** Pan-Arctic infrastructure hazard map with close-ups from central Alaska and northwestern parts of the Russian Arctic. A consensus of three geohazard indices (settlement index, risk zonation index, and analytic hierarchy process-based index) showing hazard potential by risk level (low–high) for infrastructure damage by the middle of the century (2041–2060). Hot spots indicate areas where all three indices showed high potential for infrastructure damage. Infrastructure other than settlements, the Qinghai–Tibet Railway, and major pipeline systems are shown only in the close-ups. OpenStreetMap data extracts[47,50] are licensed under CC BY-SA 2.0 (https://creativecommons.org/licenses/by-sa/2.0/) and Open Database License v1.0 (http://opendatacommons.org/licenses/odbl/1.0/). World Borders dataset is from http://thematicmapping.org/downloads/world_borders.php and licensed under CC BY-SA 3.0 (https://creativecommons.org/licenses/by-sa/3.0/)

industrial infrastructure is located in areas with high potential for near-surface permafrost thaw by 2050. Consideration of ground properties in addition to permafrost thaw showed that 33% of infrastructure is located in areas where ground subsidence and loss of structural bearing capacity could severely damage the integrity of infrastructure. The threat to hydrocarbon extraction and transportation in the Russian Arctic has been given particular emphasis. Results indicate that reducing GHG emissions and stabilizing atmospheric concentrations, under a scenario consistent with the Paris Agreement, could stabilize risks to infrastructure after mid-century. In contrast, higher GHG levels would probably result in continued detrimental climate change impacts on the built environment and economic activity in the Arctic.

## Methods

**Study design**. In-situ observations of MAGT and active layer (seasonally thawed surface layer atop permafrost) thickness (ALT), geospatial environmental data, and a statistically based ensemble method were used to model the current (2000–2014) and future MAGT and ALT in the land areas north of 30 °N degrees latitude at 30 arc-second resolution[23] (Supplementary Fig. 2). Whereas the ground thermal regime has commonly been examined using mechanistic transient models[34], an approach based on statistical associations between dependent and predictor variables was used here[22,23]. This is because statistical models are computationally more cost-efficient than mechanistic models (which currently have limited high-

resolution applicability in hemisphere scale investigations), and can account for variables related to topography and land cover that could be difficult to otherwise parametrize. Moreover, we focused on near-surface permafrost conditions (<15 m depth). Near-surface ground temperatures are strongly coupled with average atmospheric conditions, and are likely to adapt to prevailing climate conditions within a few years[35]. Cold permafrost (<−5 °C) sites and sites with low ice content and thin organic material layer normally have no substantial time lag between atmospheric forcing and ground temperature response at these depths[36]. On the contrary, ice-rich sites close to the melting point can experience a more substantial time lag; however, these sites have high potential for ground subsidence regardless of temperature development because of an abundance of excess ice in the soil[37]. More information on the strengths and weaknesses of statistical techniques in analysing permafrost in a changing climate can be found in refs [22,23].

Described below are: first, data for the modelling of MAGT and ALT, and infrastructure information; second, statistical analyses of MAGT and ALT; third, development and preparation of geohazard indices; and fourth, infrastructure hazard computations. Data and methods related to MAGT and ALT modelling are presented briefly and more information is presented in ref. [23].

**Ground temperature and active layer data**. Standardized observations of MAGT ($n = 797$) and ALT ($n = 303$) were compiled from the Global Terrestrial Network for Permafrost (GTN-P) database (gtnpdatabase.org)[38] and additional datasets for a recent 15-year period (2000–2014). MAGT observations at or near (the closest to) the depth of zero annual amplitude (ZAA, annual temperature variation < 0.1 °C)[3] were utilized. Whenever it was evident from source data or after a careful examination of the borehole location that a disturbance (e.g., the effect of geothermal heat on temperature−depth curves, recent forest fire, large water body, or

anthropogenic heat source) had an effect on the ground thermal regime, the observation was excluded.

**Geospatial environmental data**. Physically relevant climate, ground material, water body, and topographic environmental (i.e., predictor) variables were used in the statistical analysis of MAGT and ALT[23]. All the data layers were sampled to the same spatial resolution (30 arc-seconds) and extent (north of 30 °N latitude). The high-resolution WorldClim data[39] with a temporal adjustment[23] was used to compute freezing (FDD) and thawing (TDD) indices (°C days), and precipitation (mm) in solid (precipitation sum for months below 0 °C; $Prec_{T \leq 0\,°C}$) and liquid form (precipitation sum for months above 0 °C; $Prec_{T > 0\,°C}$). The creation and accuracy of the interpolated climate data are fully described in ref. [39]. In brief, weather stations ($n = 24{,}542$) behind the data have relatively even spatial coverage (excluding Greenland) and the temperature and precipitation records have passed a quality control scheme. The production of the climate surfaces is based on spline interpolation where the spatial variation in average air temperature and precipitation sums were modelled as a function of latitude, longitude, and elevation. In general, the errors between the observed and the interpolated values were small, <0.3 °C for air temperature and mostly <5 mm for precipitation, when averaged over 12 months. Climate data for future conditions are based on downscaling of multiple global climate models (GCM) from the Climate Model Intercomparison Project Phase 5 database. The GCM outputs (15 models) have been downscaled and bias-corrected for several emission scenarios (RCP2.6, RCP4.5 and RCP8.5) using the WorldClim data for current conditions as a baseline. The GCM data used here are available alongside the data for baseline conditions in ref. [39]. To control for inter-model variability in the analyses, ensemble averages over the GCM output were used for each time step and RCP scenario. Owing to a lack of suitable snow cover data for the RCP scenarios, the effect of snow cover on the ground thermal regime[40] was indirectly illustrated with monthly precipitation−temperature-derived indices. To account for the effect of organic material on the ground thermal regime[37], soil organic carbon (SOC; g kg$^{-1}$) content information was included from global SoilGrids1km data[41]. A global Water Bodies product (v 4.0) at 150 m resolution[42] allowed us to determine the percentage cover of water bodies inside each 30 arc-second grid cell. Topography-derived solar radiation was computed using the NASA Shuttle Radar Topography mission (SRTM) digital elevation model (DEM) at a 30 arc-second spatial resolution[43]. ArcGIS 10.3 software was used to derive potential incoming solar radiation (PISR; MJ cm$^{-2}$ year$^{-1}$) for each grid cell[44].

**Infrastructure, population and hydrocarbon extraction fields**. OpenStreetMap (OSM) data (www.openstreetmap.org) were the main source of infrastructure information. Considering spatial accuracy, the quality of these data is comparable to commercial geodata products[45,46]. Our goal of high-resolution (<1 km) analysis in this study demanded first-order positional accuracy for all data used. We extracted, merged, and reclassified buildings, roads, railways, industrial areas, and populated settlements from national and sub-national OSM data packages provided in ESRI Shapefiles by GeoFabrik[47]. Polygon footprints of buildings were converted to point features to examine their number within hazard zones. With roads, we adopted the reclassification method used by the International Transport Forum[48] and chose only five top-tier road types. This was assumed to reduce the risk of data quality discrepancies between regions. We discarded winter roads whenever the seasonality of a road segment was evident from the data attributes. Tunnels and ruined or non-existent elements of roads, railways, and buildings were similarly excluded. Elements under construction, however, were included in all cases with the assumption that they will be in use during the hazard assessment periods. Industrial areas represent a single class in OSM land-use data, including areas and buildings used for industrial purposes (www.openstreetmap.org). Populated settlements are an excerpt from OSM places map feature[49], which consists of a hierarchical set of locations with name and accompanying attributes. We included all types of populated settlements, encompassing isolated clusters of only a few houses to large cities.

The WikiProject Oil and Gas infrastructure served as a source for information about oil and gas pipelines[50]. The coverage and accuracy of pipeline data may vary regionally but we are not aware of any more consistent publicly available global product with comparably high spatial accuracy. In addition to OSM data, we included aviation transportation infrastructure, with ongoing global updating of data on locations of airports and airfields (OurAirports.com). Based on data attributes, currently inactive airports and seaplane bases were excluded. Human population for the year 2015 was determined using the Gridded Population of the World (GPW version 4) data in 30 arc-second resolution[51]. Population count estimates were based on national population censuses, which have been further adjusted to match 2015 Revision of United Nation World Population Prospects country totals[51]. As we are not aware of any population projection that would match with the extent of our study area, resolution, and periods analysed, we consider that using current population counts is the safest way to address human exposure to future hazards, even though changes in population, as well as in infrastructure, are probable but subject to (unpredictable) near-future socio-economic development. Hydrocarbon extraction fields in the Russian Federation

were extracted in polygon shapefiles from Rosnedra's online map interface (gis. sobr.geosys.ru). Prior to hazard computations spatially overlapping polygons were merged.

Geospatial data quality encompasses many aspects, e.g. location accuracy, completeness of elements or their attributes, which in the case of OSM have been extensively studied predominantly in highly developed areas[45,46], whereas in remote regions these evaluations are scarce. The circumpolar applicability of OSM road and railway data has been demonstrated by their previous use in the production of global 100-m resolution grids for socio-economic/population (WorldPop Project[52]) and global travel time to cities mapping at 1 km resolution[53]. Reference [54] estimated that in 2015 the global OSM road network was ~83% complete albeit between-country differences were identified. Here, we included only five top-tier road types, as opposed to smaller roads included in their analysis[54], which was assumed to reduce the risk of data quality discrepancies between regions[48]. Apart from roads, very few global-scale evaluations of the OSM data have been performed. Moreover, no systematic framework to evaluate OSM data yet exists[55].

According to our calculations, the total length of WikiProject pipelines in Russia (baseline permafrost conditions) is ~5% greater than those in the federal Rosnedra database (gis.sobr.geosys.ru). This is attributed to higher spatial resolution and a more detailed presentation of pipeline networks within communities and oil/gas fields. Compared to the documented lengths of a few central pipeline systems (including non-permafrost areas), the data encompass 99.8% of TAPS (1285/1288 km, akpipelinesafety.org), 98.9% of ESPO (4702/4756 km, energybase.ru), 93.1% of Urengoy–Pomary–Uzhgorod pipeline (4142/4451 km) and 76.2% of Bovanenkovo–Ukhta–Torzhok (2009/2637 km), suggesting that they are geospatially mostly complete and accurate.

The analyses involving buildings presented here are preliminary, as the number of OSM buildings across our modelled permafrost domain was obviously much less than the actual number. Moreover, region-specific differences exist. A simple people-per-building -ratio (regional population divided by number of buildings) was calculated to provide a rough estimate of the validity of building counts in the geographical regions under consideration. Eurasia and North America had reasonable ratios, 23 and 12, respectively, while for central Asian mountains a ratio of nearly 700 indicated that a large number of buildings could be missing. Urban settlements, which contain the majority of buildings, had good coverage, while in some of the smaller populated places infrastructure may not have been mapped. In the context of this study, which includes all settlements ranging from isolated dwellings to cities, it is important to take into account the maximum extent of human activities. This was achieved with the OSM places map feature, which includes ca. 10 times more populated settlements than analogous open datasets (e.g., the Global Rural Urban Mapping Project, Naturalearthdata.com—Populated places) across the modelled permafrost region.

**Statistical analysis**. Observed MAGT and ALT were related to geospatial environmental variables using four statistical modelling techniques: generalized linear model[56], generalized additive model[57], generalized boosting method[58] and random forest[59]. The detailed information of the model calibration and evaluation are provided in ref. [23].

The uncertainty of the model predictions (present and future) was assessed by producing 1000 ensemble predictions on 100,000 randomly chosen grid cells (glaciers masked out), at each time using a 70% random sample of observations[60]. The 95% prediction intervals for each cell over the 1000 replications were then calculated. The final uncertainty was considered as the 95th percentile of the prediction intervals across all 100,000 locations[60]. This procedure resulted in baseline prediction uncertainty of ±0.77 °C (MAGT) and ±37 cm (ALT) (Supplementary Fig. 5). For ALT, the spatial domain of the uncertainty analysis was limited to the modelled permafrost areas (i.e., MAGT ≤ 0 °C).

In the baseline period, permafrost was modelled to affect $15.1 \pm 2.1 \times 10^{-6}$ km$^2$ (95% uncertainty range) and decreased by 39.5% to $9.1 \times 10^{-6}$ km$^2$ ($7.5–11.2 \times 10^{-6}$ km$^2$) by the middle of the century[61]. These results are comparable with those presented recently[62–64]. However, an explicit comparison of the results of this study and the previous studies is difficult because of the differences in the spatial resolution of analyses (our ~1 km vs. common >100 km), extent of the study domain (e.g., our circumpolar vs. regional studies in Alaska, Siberia, Arctic Canada and Tibetan Plateau) and differences in basic settings in the analyses (e.g., depth of soil column considered, input parameters and baseline/projection periods).

**Geohazard indices**. We formulated a total of four indices (settlement index, risk zonation index, analytic hierarchy process (AHP) based index and a consensus of the former indices) depicting zones of hazard potential for infrastructure for periods 2041–2060 and 2061–2080 under three RCPs. First, the settlement index ($I_s$) was computed using the formula[8]:

$$I_s = \Delta Z_{ALT} \times V_{ice}, \qquad (1)$$

where $\Delta Z_{ALT}$ is the relative increase of ALT, and $V_{ice}$ is the volumetric proportion of excess ground-ice. We used volumetric ground-ice content (GIC) data[65] and ALT modelling results[23]. Prior to index formulation, gridded (12.5 km resolution)

GIC data were re-projected and class-specific values (5, 15 and 35%) were assigned for the original class intervals (0–10, 10–20, and over 20%). Resulting $I_s$ values were logarithmically transformed, and then reclassified into three classes using a nested-means procedure[66] with the two lower classes combined for a conservative estimate[8,67].

Second, the risk zonation index ($I_r$)[20] was developed using two-class data on surface properties (sediment/bedrock), frost susceptibility of ground material (high/low), ground-ice (high/low) and permafrost thaw potential (high/low). Assuming that permafrost thaw has a minor effect on exposed bedrock in the context of engineering, bedrock was directly assigned to the low-risk class[20]. To delimit exposed bedrock areas at 30 arc-second resolution, we used data on global soil thickness[68]. We used SoilGrids1km data[41] to produce a variable separating coarse and fine sediments with varying frost susceptibility (high susceptibility = silt and finer particles, low susceptibility = sand and coarser particles). GIC[65] was used to determine high (10–20% or more) and low (0–10%) ice content. Reference [20] developed a concept of permafrost thaw potential to describe the active layer depth increase between the present time and a future scenario (high > 2.5 m ≥ low). Owing to the greater uncertainties in ALT than in MAGT results[23], we determined permafrost thaw potential using MAGT predictions (high potential = MAGT changed from ≤0 °C to >0 °C at the depth of ZAA; low potential = MAGT remained ≤0 °C at the depth of ZAA). To compile a three-class risk zonation (comparable to the other indices) we followed the decision flow diagram[20] but merged the two classes of lowest risk (low risk and limited risk) together.

Third, AHP was used to produce a hazard index ($I_a$) that considers different factors with varying weights. The AHP developed is a multi-objective, multi-criteria decision-making approach to analyse complicated problems such as the determination of relative roles of factors affecting geohazards[69,70]. AHP requires the creation of a reciprocal pair-wise comparison matrix used to determine weighted coefficients for the computation of geohazard index. Entries into the matrix are determined by comparing each factor based on a 9-point rating scale[71]. If the factors are of equal importance for the final solution, a value of 1 is given, whereas 9 expresses the extreme importance of one criterion over another[72].

We considered the following five factors in our circumpolar-scale AHP analysis: relative increase of ALT[23]; GIC; ground temperature (including permafrost thaw); fine-grained sediment content in the ground; and slope gradient. The relative importance of each variable was ranked using expert knowledge[72] (see e.g., refs [21] and [70] for the application of AHP in geohazard context). Here, the ground temperature and thaw of near-surface permafrost was considered to be the most important factor affecting infrastructure hazard followed by GIC, relative increase of ALT, fine-grained sediment content and slope gradient. Using the expert judgement on the relative importance of factors and a reciprocal pair-wise comparison matrix we computed weighted coefficients for each factor (the resulting coefficients are shown in Eq. 2). However, the pair-wise comparison is subjective and the quality of the results is dependent on the expert's judgement. To evaluate expert valuation, a consistency ratio was used to show the probability that the assessment matrix was randomly generated[69]. In a successful expert judgement, the consistency ratio should be ≤0.1 (ref. [69]). In our AHP, the consistency ratio was 0.09, indicating acceptable assessment.

To compute $I_a$ the ground temperature, GIC, ALT, fine sediment content, and slope gradient variables were first classified into three classes (note that the GIC was originally a three-class factor) based on their corresponding contribution to infrastructure hazard in permafrost-controlled environment[21] (3 = high hazard, 2 = moderate hazard, 1 = low hazard). The ground temperature factor was produced by reclassifying the MAGT predictions. The highest hazard value was assigned to areas where near-surface permafrost thaws (comparable to the permafrost thaw potential above). The areas where MAGT stays between –3 °C and 0 °C were considered moderate hazard areas, whereas areas with MAGT < –3 °C represented the lowest hazard level[73]. Due to the lack of applicable hazard-related threshold values the nested-means approach[66] was utilized to classify the numerically continuous ALT, fine-grained sediment and slope variables into three-class factors. Fine-grained sediment content was derived from the SoilGrids1km[41] and slope gradient from the 30 arc-second DEM[43]. The hazard potential of ALT, fine sediment and slope factors was considered to increase with increasing thaw depth, fine-grained sediment content, and slope inclination, respectively. We used the coefficients and three-class raster layers to compute the AHP-based geohazard index ($I_a$):

$$I_a = (\text{ground temperature} \times 0.525) + (\text{GIC} \times 0.248) + (\text{relative increase of ALT} \times 0.122) + (\text{fine} - \text{grained sediment content} \times 0.071) + (\text{slope gradient} \times 0.035).$$

(2)

The value of $I_a$ ranged from 1.0 (lowest hazard potential) to 3.0 (highest hazard potential). To achieve a three-fold classification comparable to the $I_s$ and $I_r$, we used the nested-means procedure[66]. Finally, due to the different strengths and weaknesses of the $I_s$, $I_r$ and $I_a$ we computed a consensus index ($I_c$) using information from all the three indices. For example, $I_s$ considers two important factors causing ground subsidence in permafrost areas. However, $I_s$ does not include information on permafrost thaw or other potential hazard-inducing factors and in our study, $I_s$ was based on relatively coarse-scaled GIC data. In contrast, $I_a$ considers different affecting factors and introduces case-specific expert knowledge into the process but represents a relatively complex and subjective approach in the

development of an index for infrastructure hazard assessment. Thus, $I_c$ was computed using a majority vote procedure with ArcMap's Cell Statistics tool. Whenever two or three of the indices shared a hazard value in a grid cell, the value was recorded to represent consensus. In draw situations, i.e., all three indices had different values, a moderate hazard value of 2 is forced manually. As a result, the high hazard zone comprised 13.8%, medium hazard 41.3% and low hazard 44.9% of the study area in a medium stabilization scenario RCP4.5 (ref. [25]) for 2041–2060.

Basically, the comparison of spatial patterns of our geohazard results[61] ($I_s$, $I_r$, $I_a$, and $I_c$) to the previous studies[17,19–21,67] is challenging owing to the geographical, scale, and methodological differences between the studies (see above). The main patterns of our indices and the results published in refs [17,]67 are comparable, although local differences exist.

**Infrastructure hazard computations**. We computed the amount of infrastructure, hydrocarbon extraction fields and population in areas of permafrost thaw and hazard zones (high, moderate and low in $I_c$) using ArcGIS 10.3. First, we partitioned the infrastructure data into four geographical entities: the whole permafrost area and its extracted subsets of Eurasia, North America, and central Asian mountains. Next, *Add Geometry Attributes* script was implemented to determine the length of roads, railways, and pipelines along with the cover of industrial areas and hydrocarbon extraction fields in the permafrost thaw areas and hazard zones. In the case of point features (buildings, settlements, and airports/-fields) we used the *Multi values to points* tool in the computations. We implemented a gridded dataset of population count for the year 2015 to quantify population in anticipated hazard areas[51]. For each scenario, respective populations i.e., sums of grid cell values within areas of thawing permafrost and each hazard zone were summarized with the *Zonal statistics as a Table* tool. Similar computations were performed to consider the uncertainty in MAGT and ALT predictions (see Fig. 2, Supplementary Fig. 3).

## Data availability

The authors declare that the data supporting the findings of this study are available within the paper and its supplementary information files. All the data produced in this study are available from the corresponding author upon reasonable request or at PANGAEA, Data Publisher for Earth and Environmental Science (https://doi.pangaea.de/10.1594/PANGAEA.893881).

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

## Acknowledgements

This research was funded by the Academy of Finland grants no. 285040 and 286950 (INFRAHAZARD Consortium).

## Author contributions

M.L. developed the original idea in collaboration with J.H. J.H., O.K., M.L. J.A., S.W., V.E.R, B.E., and F.E.N. contributed to the study design. O.K. led the data compilation and infrastructure hazard computations and J.A. statistical analyses with contribution from

J.H., M.L. and O.K. J.H. led the preparation of the manuscript with contribution from O.K., J.A., M.L. S.W., F.E.N., V.E.R, and B.E.

## Additional information

**Competing interests:** The authors declare no competing interests.

