## [Peer Review File · Nature Communications]

Reviewers' comments:

Reviewer #1 (Remarks to the Author):

This manuscript focuses on a very important issue regarding thermal hazard caused by permafrost degradation under a warming climate. The thermal hazard is closely concerned by scientists and governances due to its damages to infrastructure. The issue is also one of the most important questions about the impacts of permafrost thaw, except for permafrost carbon-climate feedback, water resource, energy budget, and so on.

This study used comprehensive datasets (soil temperature, active layer, infrastructure, etc.) and statistical models to quantify the amount of infrastructure at risk due to permafrost degradation. The study is systematic and a lot of time expense. The results are attractive despite a less clear uncertainties.

In spite of these above merits, there are two other related manuscripts submitted and I see the same figure in the two manuscripts, which reduces the innovativeness and contribution of this manuscript.

Several other comments are outlined as the following:

1. The manuscript give the uncertainties of simulation results of ground temperature and active layer, but the uncertainties of the proportion of infrastructure at risk is not provided. The latter could be very pivotal for evaluation of thermal hazard by governances.
2. The manuscript does not provide the information of climate data under the RCP scenarios. For example, data source and accuracy, which are directly related to the results of future thermal hazard.
3. How are the statistical models used to identify permafrost extent? I do not find the information in the manuscript
4. For equation (1) and (2), why are these forecast factors selected? And are there other factors being not selected?
5. Extended data: Figure 5, large differences are seen in the results from four geohazard indices, especially for settlement index, Why?
6. Whether or not the present results are comparable or different with previous relevant studies, for instance, "Climate change and hazard zonation in the circum-Arctic permafrost regions, 2002, *Hat. Hazards*"; "Permafrost and changing climate: the Russian perspective, 2006, *AMBIO*"; "Permafrost thaw and associated settlement hazard onset timing over the Qinghai-Tibet engineering corridor, 2015, *Int. J. Disaster. Risk Sci.*"; "Permafrost degradation and associated ground settlement estimation under 2 °C global warming, 2016, *Clim. Dyn.*", and so on.

Reviewer #2 (Remarks to the Author):

In this manuscript, statistical models were used to predict the permafrost extent in northern Hemisphere. Meanwhile, MAGT, ALT, ground ice, soil property and slope were used to formulate geohazard indices to depict the damage caused by permafrost degradation. The viewpoints are novel and the conclusions are convincing. The results are helpful for site-specific engineering, design, and construction practices in permafrost regions.

As mentioned in the manuscript, we do not know precisely how permafrost responds to changing air temperatures in different environmental settings. The spatial and temporal responses are likely indirect, owing to the complexity of permafrost environments, especially in the marginal permafrost regions. Also, soil water content and vegetation may change with the degradation of permafrost, which may result in more uncertainty in permafrost modelling. Permafrost prediction based on statistical models may overestimate the permafrost degradation.

Ground temperature, GIC, ALT, fine sediment content, and slope gradient are used to compute Ia. In most cases, GIC, ALT, and fine sediment content are interrelated. The relative increase of ALT may exaggerate the geohazard index near the polar region where ALT is relatively smaller.

Reviewer #3 (Remarks to the Author):

It is with great interest that I have read and reviewed the submitted paper titled "Degrading permafrost puts Arctic infrastructure at risk by mid-century". The authors present a very thorough and well founded quantitative analysis of the impact of climate change on infrastructure hazards in Arctic permafrost areas.

To my knowledge, this study is the first to quantify the different types of infrastructure at risk on a circum-Arctic scale under different climate change scenarios. By quantifying infrastructure hazards at this scale, the study provides decision makers at all levels with a valuable tool in the evaluation and prioritizing of adaptation and mitigation measures.

The study is very thorough in its design at all levels:

The forecasts of permafrost change is an ensemble average of four independent statistical methods applied to the prediction of mean annual ground temperature (MAGT) and active layer thickness (ALT).

Similarly, the hazard mapping is based on three different hazard evaluation models, the results of which are consolidated through a majority-vote approach in a final consensus index to be used in the quantification of infrastructure at risk.

The data basis for both statistical modelling and hazard mapping is harvested from well-established, international data networks, such as the GTN-P, combined with local and national sources, and the process is clearly described and data sources and types listed in supplementary tables.

It is my evaluation that both methodology and the data basis are valid and of high quality, and that the study is highly relevant and recommendable for publication(s).

I would like to point out, that the submission received consists of three separate papers:

- (1) A manuscript submitted to Geophysical Research Letters.
- (2) A manuscript of unknown status (complete?, submitted?)
- (3) The manuscript submitted for review (including supplementary text and table).

Manuscript (1) describes the statistical modelling of permafrost response under different scenarios.

Manuscript (2) reproduces much of the description of the statistical modelling from manuscript (1) and adds the development/implementation of the hazard indices.

Manuscript (3) reproduces much of the descriptions (mainly in supplementary material) of manuscripts (1) and (2) and adds the quantification of infrastructure in different hazard categories based on results developed in (1) and (2).

Although these works are obviously derivative, I did not notice any cross references between them.

It is my opinion, that each paper in the sequence would gain significantly from a tighter focus on the main contribution and a discussion of the qualities of the derived product, while only providing a short description of the data products used as inputs and referencing the sources.

If such a strategy is to be followed, the manuscript reviewed here (3), should focus on the harvesting of the infrastructure data, and the analysis of infrastructure hazards, while referencing the other two papers as the source for permafrost forecasts and hazard zonation.

Manuscript (3) does have this focus in the main text, but the supplementary text (methods) venture to encompass the full range of the study, and does so with apparently less detail and discussion than (and without reference to) the original manuscripts. I find this problematic.

Specific comments to the manuscript (3) including supplementary materials:

The analysis in the paper is split in two categories "Areas of near-surface permafrost thaw" and a "High-hazard zone". This division of the analysis must be more clearly scoped. I suggest that the two classifications are briefly introduced in the introduction (i.e. before the section starting p4 L70).

- It must be clearly and concisely stated in the main paper how "permafrost thaw" is evaluated. Is it simply all grid cells that show an increase in ALT? is a threshold value used?
- Likewise it must be clearly and concisely stated in the main paper how the "high-hazard zone" is defined. The description on page 5 lines 88 to 90 is quite general and vague. Is it the grid cells where the consensus index evaluates to "high"? Or is it the hot-spot areas (where all three indices evaluate to "high")?
- I suggest to specifically state that the "high-hazard zone" is a sub-set of the "permafrost thaw" zone.

Some of the main conclusions about the effects of different climate change scenarios are made based on a comparison of means (fig 4b and extended data fig 6b). The corresponding 95% confidence intervals are quite large - have tests been made to document which level of significance can be attributed to the differences?

Page 15, Fig 2a: Please chose more contrasting colors for the two depicted permafrost zones.

Page 15, Fig 2c: The resolution in the review copy is inadequate. In addition, the choice of colors makes it impossible to distinguish the two data classes which are compared. This figure is identical to figure 2a in manuscript (1), please reference.

Page 16, Fig 3: The captions seems to mention three zoomed inserts (central Alaska, northweat Ural, Yamal-Nenets), but the figure has only two. Reformulate figure caption?

Page 17, Fig 4: This figure is very complex, and not adequately annotated/described in the caption. The figure illustrates the main conclusions of the manuscript and care must be taken to make it easily accessible. The explanation of the subfigures is insufficient.

- 1) Caption should clearly state that the subfigures show the percentage of all registered infrastructure that is located in the "permafrost thaw" zone (subfigs a+c) and the "high-hazard" zone (subfigs b+d), as a function of the different infrastructure classes (subfigs a,b,c,d), as a function of different RCP forcing (subfigs a+b), and as a function of geographical region (subfigs c+d).
- 2) How are uncertainty ranges determined for subfigures b and d? (a+c has been explained in existing caption)
- 3) Consider adding a text label on the y-axis of subfigures a and c ("Proportion of registered

infrastructure [%]"

4) Consider adding a "column" title for each set of plots "near-surface pf thaw" (a+c) and "high hazard" (b+d).

These comments apply also to extended data figure 6.

Page 21 L374-386: This section is unclear, should be reformulated. WorldClim data, Global Meteorological Forcing Dataset, Forcing data, Global forcing data, Please be consistent with terminology and specific in the description of what is done to which dataset.

Page 23 L415: "includes all types of populated settlements..." This does not seem to be the case for the map in fig 3. If a threshold value is used for plotting settlements on the map, it should be specified in the figure caption.

Page 24 Eq. 1 and 2: I am not familiar with standard notations for this type of statistical modelling, but it seems inappropriate to simply add the parameters (TDD, FDD, SOC, etc.). These have been defined in the preceding text as physical and environmental parameters, quantifiable and with specified units (e.g., deg*days for TDD and FDD, g/kg for SOC etc.). Could the equations be adapted to make it clear that the statistical model used is a function of these predictor variables? (and would the function be different for the four different statistical models used?)

Page 39 Fig 3a: Only the GAM plot has a lower whisker - why?

Page 39 Fig 3c: This plot seems almost identical to Manuscript (1) figure 2b - but statistics are somewhat different, why?

Page 42 Fig 6: See comments above for Page 17 Fig 4.

Reviewer #4 (Remarks to the Author):

Thank you for giving me the opportunity to review this interesting manuscript. Below, I provide a number of misc. comments for your consideration:

*Lines 53-54: Please provide a reference citing the relatively short lifespan of Arctic infrastructure

*Lines 83-85: At least one of these pipelines was designed with the use of thermosyphons--to mitigate some or all of the impact of thawing permafrost. Is this technology or other technologies accounted for in your analysis? It appears that a major shortcoming may be the fact that they assume that communities do not adapt to some (or all) of these changes.

*Lines 133-135: Damage to critical energy delivery and industrial infrastructure can also impact broader economic activity and national security.

*Reference #13: I do not think it is appropriate to include a reference to a newspaper article--cite the original study if it has been independently peer-reviewed.

*Little or no discussion was provided on how accurate the infrastructure count was across the Arctic (compared to other studies that have considered infrastructure at risk to permafrost degradation). For example, a study by Larsen et al. (2008)--Global Environmental Change--indicated that there was considerable uncertainty about the count and location of infrastructure located on permafrost across the U.S. state of Alaska. That study's infrastructure database was built using a bottoms-up data collection approach.

*My sense is that the uncertainty in the location, type, value and amount of infrastructure is larger than the uncertainty in the modeling of the permafrost under different climate scenarios. This

statement is consistent with other studies which point to the uncertainty in socio-economic variables exceeding the uncertainty in the geophysical variables.

*This analysis did not appear to account for population changes between now and the middle-of-the-century. We know that some places in the rural Arctic are reporting decreasing populations over time--despite overall population growth globally projected into the middle-of-the-century.

*Finally, the above implies that the authors performed a rigorous analysis on the future state of the climate/permafrost, but did not consider future population or changes in infrastructure over the next 30-40 years. These changes may be considerable--yet they do not appear to have been accounted for in this analysis.

Replies to reviewers' comments on manuscript NCOMMS-18-06236 "Degrading permafrost puts Arctic infrastructure at risk by mid-century"

Our response (**R**) to the **Reviewer** comments appear below. Line numbering in the responses refer to the PDF version of the revised manuscript (not the Word document with Track Changes).

Reviewer #1 (Remarks to the Author):

Reviewer #1: This manuscript focuses on a very important issue regarding thermal hazard caused by permafrost degradation under a warming climate. The thermal hazard is closely concerned by scientists and governances due to its damages to infrastructure. The issue is also one of the most important questions about the impacts of permafrost thaw, except for permafrost carbon-climate feedback, water resource, energy budget, and so on.

This study used comprehensive datasets (soil temperature, active layer, infrastructure, etc.) and statistical models to quantify the amount of infrastructure at risk due to permafrost degradation. The study is systematic and a lot of time expense. The results are attractive despite a less clear uncertainties.

R: Thank you for the overall positive comments on our manuscript.

Reviewer #1: In spite of these above merits, there are two other related manuscripts submitted and I see the same figure in the two manuscripts, which reduces the innovativeness and contribution of this manuscript.

R: We suppose you refer to the subfigure 2c. This particular subfigure was removed and the information content and new additional data were shown in two different subfigures. Owing to the changes we also modified the caption (see revised caption below). In general, we made substantial changes throughout the manuscript to remove overlaps with Aalto et al. (*In press*, Ref. 18 in the revised manuscript). Please also see the responses to the remarks of Reviewer #3.

(18) Aalto, J., Karjalainen, O., Hjort, J. & M. Luoto. Statistical forecasting of current and future circum-Arctic ground temperatures and active layer thickness. *Geophys. Res. Lett.* (In press). doi: [10.1029/2018GL078007](https://doi.org/10.1029/2018GL078007)

Figure 2. Distribution of permafrost in the baseline (2000–2014) and future (Representative Concentration Pathway, RCP4.5 2041–2060) climates¹⁸ (a). Note that the baseline extent of permafrost (blue) includes future distribution (greenish). The location and observed mean annual ground temperature (MAGT) of the data points (boreholes) are shown with coloured circles. Comparison of the observed and predicted MAGT in the current (baseline, 2000–2014, $n = 20,000$) based on a repeated cross-validation scheme using a 500 km distance block between calibration and evaluation boreholes (b) and past climatic conditions (hindcast) for 1970–1984 (c, $n = 250$) and 1985–1999 (d, $n = 253$) (RMSE = root-mean-square error, MAE = mean absolute error between the observed and predicted MAGT)¹⁸. For comparison with the full-data results (c and d) the hindcast RMSEs for boreholes with thaw-sensitive ‘warm permafrost’ (observed MAGT = -5 – 0 °C) were 1.32 °C ($n = 81$) and 1.15 °C ($n = 76$) for the periods of 1970–1984 and 1985–1999, respectively.

Several other comments are outlined as the following:

Reviewer #1: 1. The manuscript give the uncertainties of simulation results of ground temperature and active layer, but the uncertainties of the proportion of infrastructure at risk is not provided. The latter could be very pivotal for evaluation of thermal hazard by governances.

R: Thank you for this good remark. The uncertainty information is shown graphically in Figure 3, but we realize that it is difficult to interpret the exact values from the figure. Following the comment, we compiled a new table (Extended Data Table 3) to present the missing information.

The new table:

Extended Data Table 3. Numerical results of the proportion (%) and associated uncertainty range (see Methods for details) of infrastructure in areas of near-surface permafrost thaw and high hazard (determined by a consensus of geohazard indices) based on different Representative Concentration Pathway (RCP) scenarios by 2050 (2041–2060) and 2070 (2061–2080). The results are shown for the whole pan-Arctic permafrost area (Arctic) and its extracted subsets of Eurasia, North America and central Asian mountains (Mountains) (Settl = settlements and inh. = inhabitants).

Infrastructure	Thaw			High hazard			
	Arctic	North-America	Mountains	Arctic	Eurasia		Mountains
					North-America	Mountains	
Settl._{>5,000} inh.	66.0 (78.0-46.0)	63.9 (47.8-73.7)	85.9 (46.2-94.9)	33.7 (44.0-21.3)	26.3 (18.4-47.7)	27.3 (8.3-30.6)	
Buildings	72.4 (61.7-82.1)	76.3 (65.9-85.8)	92.4 (60.9-98.3)	26.2 (17.6-51.6)	27.4 (18.4-46.1)	12.5 (4.0-15.7)	
Roads	68.6 (42.9-80.8)	61.8 (43.1-74.4)	85.9 (42.3-97.1)	28.4 (16.2-40.5)	33.1 (20.5-45.4)	22.6 (7.4-27.3)	
Railways	82.4 (59.9-94.4)	81.3 (64.2-93.5)	83.0 (22.6-99.6)	42.7 (29.2-58.3)	39.5 (30.6-55.9)	59.0 (15.6-73.8)	
Airports	51.8 (35.6-61.5)	49.1 (34.0-60.4)	66.7 (33.3-100)	27.7 (17.0-44.7)	32.4 (18.6-53.9)	0 (0-0)	
Pipelines	62.4 (41.9-80.6)	71.8 (48.5-94.2)	100 (100-100)	28.6 (19.0-57.1)	33.3 (22.2-65.3)	0 (0-0)	
Industrial areas	42.0 (30.6-57.0)	46.5 (34.1-62.9)	53.4 (9.3-53.4)	23.9 (15.5-51.4)	26.2 (17.1-46.1)	0.1 (0.0-0.1)	
Settl._{>5,000} inh.	71.2 (78.0-58.2)	68.8 (59.3-77.6)	92.4 (65.7-96.4)	36.9 (44.0-29.2)	39.8 (32.9-56.0)	30.6 (17.1-31.0)	
Buildings	76.3 (68.1-85.8)	80.0 (72.5-88.5)	96.7 (74.9-98.7)	29.4 (22.6-49.4)	30.7 (23.8-49.9)	15.3 (6.6-15.9)	
Roads	75.8 (56.8-83.3)	67.2 (55.7-77.3)	92.4 (59.3-98.5)	31.8 (23.0-43.6)	35.8 (28.2-48.7)	25.4 (12.8-27.6)	
Railways	87.4 (71.1-98.0)	85.9 (74.3-97.6)	93.7 (89.3-99.9)	44.8 (35.5-60.8)	40.9 (35.9-58.7)	68.3 (26.4-74.2)	
Airports	57.6 (45.5-66.2)	56.6 (44.3-67.9)	100 (66.7-100)	31.0 (23.7-51.7)	37.3 (28.4-67.2)	0 (0-0)	
Pipelines	69.2 (53.9-82.4)	79.6 (61.9-96.1)	100 (100-100)	32.4 (23.8-69.6)	38.1 (27.4-67.8)	13.3 (11.8-58.9)	
Industrial areas	48.2 (36.9-65.4)	53.3 (41.1-71.7)	53.4 (30.2-97.6)	24.8 (20.1-64.4)	27.1 (22.2-42.4)	0.1 (0.1-0.1)	
Settl._{>5,000} inh.	77.2 (88.0-68.1)	76.0 (68.2-82.4)	94.9 (80.9-98.6)	40.3 (46.0-34.6)	44.4 (38.5-62.4)	30.6 (23.1-31.0)	
Buildings	85.2 (75.9-88.9)	88.2 (80.5-90.5)	98.2 (88.8-99.3)	32.4 (27.4-52.1)	33.5 (28.7-52.2)	15.6 (10.9-16.0)	
Roads	81.7 (71.3-86.7)	75.4 (69.1-81.7)	97.5 (78.6-99.6)	34.3 (29.6-48.9)	38.9 (35.4-53.8)	26.5 (19.6-27.8)	
Railways	96.8 (84.1-99.9)	96.3 (84.9-99.9)	99.8 (99.5-100)	49.5 (42.2-62.3)	45.8 (40.1-60.2)	66.5 (66.5-75.1)	
Airports	62.6 (54.2-71.5)	64.2 (55.7-74.5)	100 (66.7-100)	34.9 (28.2-63.7)	42.2 (36.3-73.5)	32.3 (25.2-60.2)	
Pipelines	81.2 (68.4-84.2)	94.8 (79.3-98.1)	100 (100-100)	39.8 (30.9-76.0)	44.8 (36.3-75.7)	23.1 (12.9-77.0)	
Industrial areas	65.1 (48.0-80.3)	72.1 (53.5-83.4)	53.4 (31.6-100)	30.6 (23.4-76.8)	33.5 (25.9-77.6)	0.1 (0.1-0.1)	
Settl._{>5,000} inh.	67.0 (78.0-48.4)	65.3 (51.1-74.3)	85.2 (44.4-94.9)	34.6 (44.0-23.2)	37.5 (26.9-52.3)	27.3 (7.9-30.6)	
Buildings	74.0 (64.3-83.1)	78.1 (68.9-86.7)	92.5 (60.4-98.2)	27.6 (19.7-45.6)	28.9 (20.7-47.2)	12.6 (3.7-15.7)	
Roads	70.1 (45.6-81.3)	63.8 (46.6-75.0)	85.7 (41.6-97.4)	29.3 (17.7-41.6)	33.5 (22.5-46.2)	22.5 (7.1-27.3)	
Railways	83.0 (62.6-95.1)	82.1 (67.3-94.3)	82.8 (21.8-99.6)	43.0 (31.0-58.9)	39.9 (32.8-56.6)	58.9 (14.8-73.8)	
Airports	53.9 (38.5-62.8)	51.9 (35.8-61.3)	66.7 (33.3-100)	29.1 (19.0-45.3)	34.5 (20.6-55.9)	0 (0-0)	
Pipelines	64.7 (46.7-81.4)	74.3 (53.8-95.1)	100 (100-100)	30.1 (20.4-63.4)	35.1 (23.6-70.9)	0 (0-0)	
Industrial areas	43.9 (33.2-60.0)	48.8 (37.0-66.3)	41.3 (9.3-53.4)	24.2 (16.9-56.6)	26.6 (18.7-61.0)	0.1 (0.0-0.1)	
Settl._{>5,000} inh.	76.6 (88.0-68.0)	75.4 (67.8-82.0)	94.9 (81.9-98.6)	40.2 (46.0-34.6)	44.3 (38.3-62.3)	30.1 (26.7-61.7)	
Buildings	84.9 (75.0-89.1)	88.1 (79.5-90.3)	98.1 (89.1-99.3)	32.4 (27.3-52.1)	33.2 (28.6-52.2)	15.6 (11.4-16.0)	
Roads	81.3 (70.7-86.5)	75.0 (68.3-81.3)	97.3 (78.7-99.6)	34.1 (29.4-48.4)	38.6 (30.0-53.5)	26.5 (19.8-27.8)	
Railways	96.2 (83.3-99.9)	95.6 (83.9-99.9)	99.2 (69.6-100)	49.0 (41.7-62.3)	45.3 (39.5-60.2)	73.1 (48.7-74.5)	
Airports	62.8 (53.7-72.0)	66.0 (54.7-74.5)	100 (66.7-100)	35.5 (27.9-58.7)	44.1 (35.3-73.5)	32.3 (25.2-58.7)	
Pipelines	81.1 (66.3-84.1)	94.7 (76.6-98.0)	100 (100-100)	40.7 (29.6-75.8)	44.8 (34.7-75.7)	27.0 (12.9-76.3)	
Industrial areas	64.1 (46.4-79.7)	71.0 (51.7-83.2)	53.4 (31.6-100)	29.9 (23.2-76.2)	32.4 (25.6-77.3)	0.1 (0.1-0.1)	
Settl._{>5,000} inh.	87.5 (94.0-80.8)	87.2 (80.6-94.2)	99.3 (95.3-100)	50.3 (50.0-41.5)	56.7 (46.1-76.5)	31.0 (30.6-31.0)	
Buildings	90.7 (86.4-95.2)	92.6 (89.4-95.9)	100 (98.0-100)	37.6 (32.0-61.4)	37.4 (32.7-59.9)	15.7 (15.3-16.0)	
Roads	89.2 (84.1-93.0)	85.3 (79.4-89.4)	99.9 (96.8-100)	40.6 (35.1-61.8)	47.5 (40.8-67.7)	26.6 (26.0-27.8)	
Railways	100 (99.4-100)	100 (99.7-100)	100 (96.3-100)	51.7 (50.7-63.8)	48.4 (47.5-60.9)	73.7 (71.0-74.5)	
Airports	74.9 (66.0-86.4)	78.3 (71.7-87.7)	100 (100-100)	48.3 (33.8-85.8)	58.8 (45.1-84.3)	44.5 (29.5-87.0)	
Pipelines	86.2 (83.2-96.1)	99.8 (97.2-99.9)	100 (100-100)	51.9 (37.9-84.8)	47.3 (45.3-81.5)	67.3 (12.9-96.2)	
Industrial areas	88.1 (73.1-94.2)	93.9 (81.1-99.8)	100 (53.4-100)	47.6 (34.1-87.0)	48.2 (37.5-86.4)	44.9 (9.2-95.7)	

Moreover, we added comparable uncertainty information (in the columns “In thaw area” and “In high hazard area”) for Extended Data Table 2 (deleted text ~~erossed-out~~; added text underlined).

Extended Data Table 2. Numerical results of the hazard computations for pan-Arctic population (2015), hydrocarbon extraction fields in the Russian Arctic, major pipelines, and central railways on permafrost using a consensus index (I_c) for periods 2041–60 and 2061–80 under three Representative Concentration Pathways (RCPs) (Methods). Proportion (%) and associated uncertainty range (see Methods for details) of elements at risk in areas of near-surface permafrost thaw and high hazard (determined by a consensus of geohazard indices, I_c) appear in brackets. The areal coverage of I_c is slightly smaller than that of modelled permafrost owing to the patchiness of sediment property data in certain high-Arctic areas³⁹.

	Element at risk	On permafrost	In thaw area	in I_c area	In high hazard area
RCP2.6 2041–2060	Population	4,906,854	3,093,472 (63.0, 33.3-82.7)	4,567,438	721,308 (15.8, 9.3-30.4)
	Oil/Gas fields in the Russian Arctic	84,170	61,298 (72.8, 51.5-82.5)	83,870	35,143 (41.9, 31.9-71.2)
	TAPS (Prudhoe Bay-Valdez)	914	492 (53.9, 30.5-63.9)	913	291 (31.9, 20.9-57.2)
	Yamal-Nenets gas pipelines	1,341	1,182 (88.2, 54.6-100)	1,341	593 (44.3, 23.5-80.7)
	ESPO (Tayshet-Kozmino)	1,600	1,554 (97.2, 85.6-100)	1,600	733 (45.8, 43.6-46.6)
	Qinghai-Tibet railway (Lhasa-Xining)	510	412 (80.9, 20.6-99.6)	479	379 (79.1, 20.1-99.0)
	Obstkaya-Bovanenkovo railway	563	245 (43.4, 8.4-56.8)	563	176 (31.2, 5.8-76.0)
RCP4.5 2041–2060	Population	4,906,854	3,612,746 (73.6, 46.3-87.3)	4,567,438	944,995 (20.7, 12.4-33.4)
	Oil/Gas fields in the Russian Arctic	84,170	64,560 (76.7, 61.1-85.4)	83,870	38,004 (45.3, 34.9-77.9)
	TAPS (Prudhoe Bay-Valdez)	914	555 (60.7, 44.8-66.2)	913	293 (32.1, 29.1-66.7)
	Yamal-Nenets gas pipelines	1,341	1,262 (94.1, 69.7-100)	1,341	673 (50.2, 29.5-92.8)
	ESPO (Tayshet-Kozmino)	1,600	1,586 (99.1, 93.8-100)	1,600	733 (45.8, 45.8-46.6)
	Qinghai-Tibet railway (Lhasa-Xining)	510	473 (92.9, 36.6-99.8)	479	438 (91.4, 34.6-99.5)
	Obstkaya-Bovanenkovo railway	563	277 (49.2, 18.7-81.8)	563	209 (37.1, 13.5-91.0)
RCP8.5 2041–2060	Population	4,906,854	4,120,106 (84.0, 65.5-92.5)	4,567,438	1,061,324 (23.2, 19.7-37.2)
	Oil/Gas fields in the Russian Arctic	84,170	73,070 (86.8, 76.1-94.1)	83,870	48,143 (57.4, 44.4-84.0)
	TAPS (Prudhoe Bay-Valdez)	914	589 (64.5, 56.6-69.7)	913	314 (34.4, 31.7-76.4)
	Yamal-Nenets gas pipelines	1,341	1,341 (100, 90.2-100)	1,341	752 (56.1, 45.7-100)
	ESPO (Tayshet-Kozmino)	1,600	1,600 (100, 98.1-100)	1,600	733 (45.8, 45.8-46.6)
	Qinghai-Tibet railway (Lhasa-Xining)	510	506 (99.3, 66.8-100)	479	470 (98.0, 64.5-99.9)
	Obstkaya-Bovanenkovo railway	563	422 (74.9, 45.3-100)	563	406 (72.1, 32.9-100)
RCP2.6 2061–2080	Population	4,906,854	3,134,647 (63.9, 34.1-83.3)	4,567,438	765,476 (16.8, 9.9-31.1)
	Oil/Gas fields in the Russian Arctic	84,170	62,281 (74.0, 55.6-83.4)	83,870	35,901 (42.8, 33-76.5)
	TAPS (Prudhoe Bay-Valdez)	914	523 (57.3, 36.5-64.8)	913	292 (31.9, 24.3-62.4)
	Yamal-Nenets gas pipelines	1,341	1,188 (88.6, 59.1-100)	1,341	599 (44.7, 24.4-87.0)
	ESPO (Tayshet-Kozmino)	1,600	1,559 (97.5, 88.1-100)	1,600	733 (45.8, 44.8-46.6)
	Qinghai-Tibet railway (Lhasa-Xining)	510	411 (80.8, 19.4-99.6)	479	378 (79.0, 19.1-99.0)
	Obstkaya-Bovanenkovo railway	563	251 (44.5, 9.7-58.5)	563	183 (32.4, 6.7-77.4)
RCP4.5 2061–2080	Population	4,906,854	4,108,172 (83.7, 64.8-92.1)	4,567,438	1,052,540 (23.0, 19.4-36.9)
	Oil/Gas fields in the Russian Arctic	84,170	72,037 (85.6, 74.0-92.6)	83,870	47,162 (56.2, 42.3-83.0)
	TAPS (Prudhoe Bay-Valdez)	914	589 (64.5, 56.3-68.5)	913	318 (34.8, 31.7-74.7)
	Yamal-Nenets gas pipelines	1,341	1,341 (100, 86.5-100)	1,341	752 (56.1, 42.1-100)
	ESPO (Tayshet-Kozmino)	1,600	1,600 (100, 98.0-100)	1,600	733 (45.8, 45.8-46.6)
	Qinghai-Tibet railway (Lhasa-Xining)	510	506 (99.3, 67.3-100)	479	470 (98.0, 64.8-99.9)
	Obstkaya-Bovanenkovo railway	563	379 (67.2, 43.6-100)	563	373 (66.3, 31.2-100)
RCP8.5 2061–2080	Population	4,906,854	4,634,539 (94.5, 86.8-97.1)	4,567,438	1,287,285 (28.2, 23.6-43.7)
	Oil/Gas fields in the Russian Arctic	84,170	83,832 (99.6, 92.4-100)	83,870	57,350 (68.4, 60.2-91.4)
	TAPS (Prudhoe Bay-Valdez)	914	696 (76.2, 66.6-97.2)	913	496 (54.3, 31.7-97.9)
	Yamal-Nenets gas pipelines	1,341	1,341 (100, 100-100)	1,341	752 (56.1, 55.5-100)
	ESPO (Tayshet-Kozmino)	1,600	1,600 (100, 100-100)	1,600	741 (46.3, 45.8-49.1)
	Qinghai-Tibet railway (Lhasa-Xining)	510	510 (100, 96.5-100)	479	473 (98.8, 95.0-99.9)
	Obstkaya-Bovanenkovo railway	563	563 (100, 100-100)	563	495 (87.8, 87.1-100)

TAPS = Trans-Alaska Pipeline System; ESPO = Eastern Siberia–Pacific Ocean pipeline

Reviewer #1: 2. The manuscript does not provide the information of climate data under the RCP scenarios. For example, data source and accuracy, which are directly related to the results of future thermal hazard.

R: The production and uncertainties of the used climate data (i.e. the WorldClim dataset) are fully described in Hijmans et al. (2005). However, we agree that such information is useful for the

readers and thus we have added a brief description of the climate data to the Methods section (lines 274–288):

“The creation and accuracy of the interpolated climate data are fully described in Ref. (37). In brief, weather stations (n=24,542) behind the data have a relatively equal spatial coverage (excluding Greenland) and the temperature and precipitation records have passed a quality control scheme. The production of the climate surfaces are based on spline interpolation where the spatial variation in average air temperature and precipitation sums were modelled as a function of latitude, longitude and elevation. In general, the errors between the observed and the interpolated values were small, < 0.3°C for air temperature and mostly < 5 mm for precipitation, when averaged over 12 months. Climate data for future conditions are based on downscaling of multiple global climate models (GCM) from the Climate Model Intercomparison Project Phase 5 (CMIP5) database. The GCM outputs (15 models) have been downscaled and bias-corrected for several emission scenarios (RCP2.6, RCP4.5 and RCP8.5) using the WorldClim data for current conditions as a baseline. The GCM data used here are available alongside the data for baseline conditions in Ref. (37). To control for inter-model variability in the analyses, ensemble averages over the GCM output was used for each time step and RCP scenario.”

(37) Hijmans, R. J., Cameron, S. E., Parra, J. L., Jones, P. G. & Jarvis, A. Very high resolution interpolated climate surfaces for global land areas. *Int. J. Climatol.* **25**, 1965–1978 (2005).

Reviewer #1: 3. How are the statistical models used to identify permafrost extent? I do not find the information in the manuscript

R: We used the predictions of mean annual ground temperature (MAGT) at zero annual amplitude depth to indicate the suitable conditions for presence ($\text{MAGT} \leq 0 \text{ }^\circ\text{C}$) and absence ($\text{MAGT} > 0 \text{ }^\circ\text{C}$) of permafrost. This information is in the revised manuscript (lines 52–55) (see also the response to Reviewer #3).

Reviewer #1: 4. For equation (1) and (2), why are these forecast factors selected? And are there other factors being not selected?

R: We used permafrost literature and theoretical knowledge but also considered the quality and coverage of the potential data sources and suitability in climate change projections when selecting physically relevant environmental variables. Consideration of vegetation and land-use could locally improve the models but we excluded these variables from the analyses owing to their presumably dynamic nature and the lack of applicable projections for the future periods. However, the selection (or omission) of specific variables was not elaborated here because the details of the statistical analyses are presented in Aalto et al. (*In press*) (a reference to this paper was added).

Reviewer #1: 5. Extended data: Figure 5, large differences are seen in the results from four geohazard indices, especially for settlement index, Why?

R: All three indices (settlement index, risk zonation index, and analytic hierarchy process-based index) were based on different environmental variables in their formulations. Therefore, the indices display different hazardous conditions stemming from different processes. Acknowledging the strengths and weaknesses associated with each index, a consensus method was employed to reduce uncertainty and to detect the most hazardous areas. For the settlement index, pronouncedly high hazard values in the high-Arctic areas can be explained by the great weight given to the relative change of ALT used in the index formulation. Even minor future increases in thaw, especially in the

high-Arctic where only shallow initial annual thaw occurs and the massive ground ice is often located right below the active layer, translated into large relative changes and therefore elevated hazard potential. An interconnection between high relative ALT change and high hazard potential was similarly pronounced in the original index by Nelson et al. (2001, 2002) and in a settlement index reproduction by Anisimov and Reneva (2006). Different properties of the employed indices were briefly considered in the original submission (lines 442–448) and no further additions were conducted to keep the text concise. However, we slightly modified the figure (Extended Data Figure 2 in the revised manuscript; added text underlined) to improve the interpretability of the Arctic region.

Extended Data Figure 2. Geohazard indices showing hazard potential by risk level (low, moderate and high) for infrastructure damage by the middle of the century (RCP4.5 2041–2060). Settlement index (a), risk zonation index (b), analytic hierarchy process-based index (c), and a consensus of the three indices (d) are presented. Note that some of the mid-latitude mountains were excluded to improve the interpretability of the Arctic region.

(9) Nelson F. E., Anisimov O. A. & Shiklomanov N. I. Subsidence risk from thawing permafrost. *Nature* **410**, 889–890 (2001).

(61) Nelson, F. E., Anisimov, O. A., & Shiklomanov, N. I. Climate change and hazard zonation in the circum-Arctic permafrost regions. *Nat. Haz.* **26**, 203–225 (2002).

(70) Anisimov, O., Reneva, S. Permafrost and changing climate: the Russian perspective. *Ambio* **35**, 169–175 (2006).

Reviewer #1: 6. Whether or not the present results are comparable or different with previous relevant studies, for instance, “Climate change and hazard zonation in the circum-Arctic permafrost regions, 2002, *Hat. Hazards*”; “Permafrost and changing climate: the Russian perspective, 2006, *AMBIO*”; “Permafrost thaw and associated settlement hazard onset timing over the Qinghai-Tibet engineering corridor, 2015, *Int. J. Disaster. Risk Sci.*”; “Permafrost degradation and associated ground settlement estimation under 2 °C global warming, 2016, *Clim. Dyn.*”, and so on.

R: During the preparation of the manuscript, we went through a substantial amount of publications concerning climate change and (i) permafrost extent (e.g. McGuire et al. 2016; Chadburn et al. 2017; Guo & Wang 2017), (ii) active layer thickness (e.g. Mishra et al. 2017; Yi et al. 2018) and (iii) geohazard indices (e.g. Nelson et al. 2002; Anisimov & Reneva 2006; Daanen et al. 2011; Shiklomanov et al. 2017). Based on our literature survey, we found it challenging (and in some cases even questionable) to make direct comparisons between our results and the results presented in the literature because of: (i) the differences in the spatial resolution of analyses (our ~1 km vs. common >100 km), (ii) extent of the study domain (e.g. our circumpolar vs. more local studies in Alaska, Russian Arctic and Tibet) and (iii) differences in basic settings in the analyses (e.g. depth of soil column considered, input parameters and baseline/projection periods). Consequently, to make meaningful comparisons, the basic differences between our and the published studies should also be presented and discussed, and this would lengthen the manuscript unnecessarily and is mainly beyond the scope of this study. However, we acknowledge the need to consider the most relevant literature and to make some general conclusions on the comparability of the results. Thus, we added two short sections in the Methods. These new discussions were inserted into the Methods to keep the main text concise and reader friendly, but also because of the difficulties to make any explicit comparisons (see above) between this study and the published literature.

Lines 357–364: “In the baseline period, permafrost was modelled to affect $15.1 \pm 2.1 \times 10^{-6} \text{ km}^2$ (95% uncertainty range) and decreased by 39.5% to $9.1 \times 10^{-6} \text{ km}^2$ ($7.5\text{--}11.2 \times 10^{-6} \text{ km}^2$) by the middle of the century. These results are comparable with those presented recently^{56–58}. However, an explicit comparison of the results of this study and the previous studies is difficult because of the differences in the spatial resolution of analyses (our ~1 km vs. common >100 km), extent of the study domain (e.g. our circumpolar vs. regional studies in Alaska, Siberia, Arctic Canada and Tibetan Plateau) and (iii) differences in basic settings in the analyses (e.g. depth of soil column considered, input parameters and baseline/projection periods).”

Lines 456–459: “Basically, the comparison of spatial patterns of our geohazard results (I_s , I_r , I_a , and I_c) to the previous studies^{27,61,62,68,70} is challenging owing to the geographical, scale and methodological differences between the studies (see above). The main patterns of our indices and the results published in Refs (27) and (61) are comparable although local differences exist.”

- (27) Shiklomanov, N. I., Streletskiy, D. A., Swales, T. B. & Kokorev, V. A. Climate Change and Stability of Urban Infrastructure in Russian Permafrost Regions: Prognostic Assessment based on GCM Climate Projections. *Geog. Rev.* **107**, 125-142 (2017).
- (56) Guo, D., Wang, H. Permafrost degradation and associated ground settlement estimation under 2 C global warming. *Clim. Dyn.* **49**, 2569-2583 (2017).
- (57) McGuire, A. D. et al. Variability in the sensitivity among model simulations of permafrost and carbon dynamics in the permafrost region between 1960 and 2009. *Glob. Biogeochem. Cycles* **30**, 1015–1037 (2016).
- (58) Chadburn, S. E. et al. An observation-based constraint on permafrost loss as a function of global warming. *Nature Clim. Change* **7**, 340–344 (2017).
- (61) Nelson, F. E., Anisimov, O. A., & Shiklomanov, N. I. Climate change and hazard zonation in the circum-Arctic permafrost regions. *Nat. Haz.* **26**, 203–225 (2002).
- (62) Daanen, R. P., et al. Permafrost degradation risk zone assessment using simulation models. *The Cryosphere* **5**, 1043–1056 (2011).
- (68) Hong, E., Perkins, R. & Trainor, S. Thaw Settlement Hazard of Permafrost Related to Climate Warming in Alaska. *Arctic* **67**, 93-103 (2014).
- (70) Anisimov, O., Reneva, S. Permafrost and changing climate: the Russian perspective. *Ambio* **35**, 169–175 (2006).

Deleted or not included references:

Mishra, U., Drewniak, B., Jastrow, J. D., & Matamala, R. M.. Spatial representation of organic carbon and active-layer thickness of high latitude soils in CMIP5 earth system models. *Geoderma* **300**, 55–63 (2017).

Yi, Y., et al. Characterizing permafrost active layer dynamics and sensitivity to landscape spatial heterogeneity in Alaska. *The Cryosphere* **12**, 145–161 (2018).

Reviewer #2 (Remarks to the Author):

Reviewer #2: In this manuscript, statistical models were used to predict the permafrost extent in northern Hemisphere. Meanwhile, MAGT, ALT, ground ice, soil property and slope were used to formulate geohazard indices to depict the damage caused by permafrost degradation. The viewpoints are novel and the conclusions are convincing. The results are helpful for site-specific engineering, design, and construction practices in permafrost regions.

R: Thank you for the positive and encouraging comment.

Reviewer #2: As mentioned in the manuscript, we do not know precisely how permafrost responds to changing air temperatures in different environmental settings. The spatial and temporal responses are likely indirect, owing to the complexity of permafrost environments, especially in the marginal permafrost regions. Also, soil water content and vegetation may change with the degradation of

permafrost, which may result in more uncertainty in permafrost modelling. Permafrost prediction based on statistical models may overestimate the permafrost degradation.

R: This is a very important comment. It is true that statistical models are static and this feature probably increase uncertainty in climate change explorations (e.g. response of permafrost to changing conditions). As in this study such problem can be partly controlled by using observations that cover investigated environmental gradients and by applying ensembles of different statistical techniques each with different statistical assumptions. Moreover, we focused on near-surface permafrost. Near-surface ground temperatures are strongly coupled with average atmospheric conditions, and are often characterized by cumulative temperature sums (e.g. freezing and thawing degree days) as done in this study. For example, ground temperatures at the zero annual amplitude depth are likely to adapt to prevailing climate conditions within few years (e.g. Streletskiy et al. 2015). Still, several physical issues that cannot be fully addressed with statistical modelling may complicate the ground thermal response to a changing climate. To our opinion, we considered these issues concisely in the Method section (line 237–243, 245–250). However, we added one sentence to highlight the connection between atmospheric and near-surface ground temperatures (“Near-surface ground temperatures are strongly coupled with average atmospheric conditions, and are likely to adapt to prevailing climate conditions within few years³².”) and one sentence where more information on the methodological issues can be found (“More information on the strengths and weaknesses of statistical techniques in analysing permafrost in a changing climate can be found in Refs (17) and (18).”).

Explanation for the omission of certain potential variables was given above (response to Reviewer #1).

(17) Aalto, J., Harrison, S. & Luoto, M. Statistical modelling predicts almost complete loss of major periglacial processes in Northern Europe by 2100. *Nat. Commun.* **8**, 515. (2017).

(18) Aalto, J., Karjalainen, O., Hjort, J. & Luoto, M. Statistical forecasting of current and future circum-Arctic ground temperatures and active layer thickness. *Geophys. Res. Lett.* (In press). doi.org/10.1029/2018GL078007

(32) Streletskiy, D. A., Sherstiukov, A. B., Frauenfeld, O. W. & Nelson, F. E. Changes in the 1963–2013 shallow ground thermal regime in Russian permafrost regions. *Environ. Res. Lett.* **10**, 125005 (2015).

Reviewer #2: Ground temperature, GIC, ALT, fine sediment content, and slope gradient are used to compute Ia. In most cases, GIC, ALT, and fine sediment content are interrelated. The relative increase of ALT may exaggerate the geohazard index near the polar region where ALT is relatively smaller.

R: It is true that GIC, ALT, and fine sediment content are commonly interrelated but there can also be clear differences in these variables owing to locally varying environmental conditions. To our opinion, they all highlight partly different aspects of geohazard potential in permafrost regions. Moreover, we used different data sources for these variables and a consensus method in the computation of hazard index for the infrastructure risk assessments.

Especially in the high-Arctic, where only shallow initial annual thaw occurs, even minor future increases in thaw depth translate into large relative changes and therefore elevated hazard potential. This is mostly visible in the settlement index, in the formulation of which the relative change of

ALT has a great weight. However, we wanted to use the original formulation of the settlement index (Nelson et al. 2001, 2002) (please see the response to Reviewer #1).

(9) Nelson F. E., Anisimov O. A. & Shiklomanov N. I. Subsidence risk from thawing permafrost. *Nature* **410**, 889–890 (2001).

(61) Nelson, F. E., Anisimov, O. A., & Shiklomanov, N. I. Climate change and hazard zonation in the circum-Arctic permafrost regions. *Nat. Haz.* **26**, 203–225 (2002).

Reviewer #3 (Remarks to the Author):

Reviewer #3: It is with great interest that I have read and reviewed the submitted paper titled "Degrading permafrost puts Arctic infrastructure at risk by mid-century". The authors present a very thorough and well founded quantitative analysis of the impact of climate change on infrastructure hazards in Arctic permafrost areas.

To my knowledge, this study is the first to quantify the different types of infrastructure at risk on a circum-Arctic scale under different climate change scenarios. By quantifying infrastructure hazards at this scale, the study provides decision makers at all levels with a valuable tool in the evaluation and prioritizing of adaptation and mitigation measures.

The study is very thorough in its design at all levels:

The forecasts of permafrost change is an ensemble average of four independent statistical methods applied to the prediction of mean annual ground temperature (MAGT) and active layer thickness (ALT).

Similarly, the hazard mapping is based on three different hazard evaluation models, the results of which are consolidated through a majority-vote approach in a final consensus index to be used in the quantification of infrastructure at risk.

The data basis for both statistical modelling and hazard mapping is harvested from well-established, international data networks, such as the GTN-P, combined with local and national sources, and the process is clearly described and data sources and types listed in supplementary tables.

It is my evaluation that both methodology and the data basis are valid and of high quality, and that the study is highly relevant and recommendable for publication(s).

R: Thank you very much. We are delighted to get these positive assessments.

Reviewer #3: I would like to point out, that the submission recieved consists of three separate papers:

- (1) A manuscript submitted to Geophysical Research Letters.
- (2) A manuscript of unknown status (complete?, submitted?)
- (3) The manuscript submitted for review (including supplementary text and table).

Manuscript (1) describes the statistical modelling of permafrost response under differencet scenarios. Manuscript (2) reproduces much of the description of the statistical modelling from manuscript (1) and adds the development/implementation of the hazard indices.

Manuscript (3) reproduces much of the descriptions (mainly in supplementary material) of manuscripts (1) and (2) and adds the quantification of infrastructure in different hazard categories based on results developed in (1) and (2).

Although these works are obviously derivative, I did not notice any cross references between them.

R: Based on the knowledge that manuscripts should not be used as references in *Nature Communications*, we prepared a stand-alone manuscript here because the status of the manuscript (1) (submitted to *Geophysical Research Letters, GRL*) and the other manuscript (2) (referred above as “A manuscript of unknown status”) was unsolved. Now, when the submission to *GRL* (Aalto et al., *In press*) is accepted for publication, we realize that we have to remove the overlaps. Thus, we conducted the required changes to this manuscript (please see below). We would like to highlight that these changes did not affect the main content, results or conclusions of this submission i.e. they did not compromise the innovativeness nor novelty value of this manuscript. The manuscript (2) is a data descriptor that will be modified according the result of this (*Nature Communications*) submission. More importantly, this *Nature Communications* submission is the original study for the computation of geohazard indices.

Reviewer #3: It is my opinion, that each paper in the sequence would gain significantly from a tighter focus on the main contribution and a discussion of the qualities of the derived product, while only providing a short description of the data products used as inputs and referencing the sources. If such a strategy is to be followed, the manuscript reviewed here (3), should focus on the harvesting of the infrastructure data, and the analysis of infrastructure hazards, while referencing the other two papers as the source for permafrost forecasts and hazard zonation.

R: Great thanks for your advice. We followed this logic but kept the hazard zonation part in this manuscript (please see the explanation below).

Reviewer #3: Manuscript (3) does have this focus in the main text, but the supplementary text (methods) venture to encompass the full range of the study, and does so with apparently less detail and discussion than (and without reference to) the original manuscripts. I find this problematic.

R: Thank you for supporting the content of the main text in its current form. In this revision, we followed your recommendations as much as possible considering the fact that data and methods concerning hazard zonation cannot be removed from this study (i.e. this is the original work for the development of geohazard indices). Consequently, we (i) removed majority of the data and method descriptions and results (including three Extended Data Figures) unique to the statistical modelling of MAGT and ALT from the Methods section and (ii) referred to the Aalto et al. paper published online in *GRL*. Although we could not remove the main data and method descriptions of geohazard indices and zonation, we made some changes to compress the method description. Overall, the word count of the Method section was clearly reduced (from ca. 3600 to below 3000 words).

Following the recommendations above and to ensure that the flow of the work will be followed by wide readership we kept the very basic parts of the statistical analyses but without duplicating anything that was presented in the *GRL* paper. For example, we kept the map showing the change in permafrost distribution (not considered in *GRL*) between the baseline and mid-century because this is crucial for understanding the main content of this manuscript. Considering thaw-sensitive ‘warm permafrost’ sites we made an independent hindcast analysis and reported shortly the results in this manuscript. For clarity and comparison we reported (with a reference to Aalto et al.) also the hindcast results in Fig. 2 using the full range of observations. Again, without duplicating figures presented in the *GRL* paper.

Reviewer #3: Specific comments to the manuscript (3) including supplementary materials:

The analysis in the paper is split in two categories "Areas of near-surface permafrost thaw" and a "High-hazard zone". This division of the analysis must be more clearly scoped. I suggest that the two classifications are briefly introduced in the introduction (i.e. before the section starting p4 L70).

R: We added information related to these two hazard categories (lines 62–63).

Reviewer #3: - It must be clearly and concisely stated in the main paper how "permafrost thaw" is evaluated. Is it simply all grid cells that show an increase in ALT? is a threshold value used?

R: A definition was added (lines 52–55): "Near surface permafrost was considered to thaw when mean annual ground temperature at or near (the closest to) the depth of zero annual amplitude changed from ≤ 0 °C to > 0 °C between the baseline (2000–2014) and future period."

Reviewer #3: - Likewise it must be clearly and concisely stated in the main paper how the "high-hazard zone" is defined. The description on page 5 lines 88 to 90 is quite general and vague. Is it the grid cells where the consensus index evaluates to "high"? Or is it the hot-spot areas (where all three indices evaluate to "high")?

R: The "high hazard" refers to areas where the consensus index indicated "high hazard". We consider that the specification made above (lines 62–63) covers this shortage as well.

Reviewer #3: - I suggest to specifically state that the "high-hazard zone" is a sub-set of the "permafrost thaw" zone.

R: Although this is the situation (high-hazard zone is a sub-set of the permafrost thaw zone) for majority of the analysed grid cells, this is not the case for all grid cells. A grid cell can be a high-hazard cell based on the consensus index even without thaw of near-surface permafrost [e.g. when settlement index (this index did not include information on thaw of near-surface permafrost) and analytic hierarchy process (AHP) based index indicated high-hazard]. Owing to the multivariate nature of the AHP index it could get a value of "high hazard" even if permafrost was not considered to thaw i.e. MAGT stayed between -3 °C and 0 °C). No changes were made.

Reviewer #3: Some of the main conclusions about the effects of different climate change scenarios are made based on a comparison of means (fig 4b and extended data fig 6b). The corresponding 95% confidence intervals are quite large - have tests been made to document which level of significance can be attributed to the differences?

R: We did not consider it feasible to make a statistical test based on only few "observations" (Fig. 4b). To our opinion, our conclusions (the use of words) included this uncertainty already: "...Results indicate that reducing GHG emissions and stabilizing atmospheric concentrations, under a scenario consistent with the Paris Agreement, could stabilize risks to infrastructure after mid-century. In contrast, higher GHG levels would probably result in continued detrimental climate change impacts on the built environment and economic activity in the Arctic." No changes were made.

Reviewer #3: Page 15, Fig 2a: Please chose more contrasting colors for the two depicted permafrost zones.

R: This figure was edited as suggested.

Reviewer #3: Page 15, Fig 2c: The resolution in the review copy is inadequate. In addition, the choice of colors makes it impossible to distinguish the two data classes which are compared. This figure is identical to figure 2a in manuscript (1), please reference.

R: This subfigure was removed [the information content was split to two individual panels (**c** and **d**) and a new evaluation measure (mean absolute error between the observed and predicted MAGT) was computed and included] (please see the response to Reviewer #1).

Reviewer #3: Page 16, Fig 3: The captions seems to mention three zoomed inserts (central Alaska, northweat Ural, Yamal-Nenets), but the figure has only two. Reformulate figure caption?

R: The caption was reformulated: "Figure 3. Pan-Arctic infrastructure hazard map with close-ups from the central Alaska and northwestern parts of the Russian Arctic. ..."

Reviewer #3: Page 17, Fig 4: This figure is very complex, and not adequately annotated/described in the caption. The figure illustrates the main conclusions of the manuscript and care must be taken to make it easily accessible. The explanation of the subfigures is insufficient.

1) Caption should clearly state that the subfigures show the percentage of all registered infrastructure that is located in the "permafrost thaw" zone (subfigs a+c) and the "high-hazard" zone (subfigs b+d), as a function of the different infrastructure classes (subfigs a,b,c,d), as a function of different RCP forcing (subfigs a+b), and as a function of geographical region (subfigs c+d).

R: We improved the accessibility of the figure according to the comments (see below).

Reviewer #3: 2) How are uncertainty ranges determined for subfigures b and d? (a+c has been explained in existing caption)

R: The missing information was added. For sub figures **b** and **d** we used uncertainty in the mean annual ground temperature (MAGT) and active layer thickness (ALT) predictions. For sub figures **a** and **c** we used only uncertainty in the MAGT predictions (if near-surface permafrost thaws up to the depth of zero annual amplitude, ALT becomes irrelevant).

Reviewer #3: 3) Consider adding a text label on the y-axis of subfigures a and c ("Proportion of registered infrastructure [%]")

R: Labels added.

Reviewer #3: 4) Consider adding a "column" title for each set of plots "near-surface pf thaw" (a+c) and "high hazard" (b+d).

R: Labels added.

Reviewer #3: These comments apply also to extended data figure 6.

R: All the above changes were also made to the comparable figure in the Extended Data (in the revised version Extended Data Figure 3)

Reviewer #3: Page 21 L374-386: This section is unclear, should be reformulated. WorldClim data, Global Meteorological Forcing Dataset, Forcing data, Global forcing data, Please be consistent with terminology and specific in the description of what is done to which dataset.

R: This section was removed during the revision (please see above).

Reviewer #3: Page 23 L415: "includes all types of populated settlements..." This does not seem to be the case for the map in fig 3. If a threshold value is used for plotting settlements on the map, it should be specified in the figure caption.

R: Figure 3 includes all the populated settlements that were included in the data (OpenStreetMap) and were on our modelled permafrost area (i.e. no threshold was determined to exclude settlements). Some coast margins (and potentially few settlements) were cut out during the merging of different data from different sources (e.g. the original resolution of ground ice content data was rather coarse, 12.5 km). In mid-latitude mountains, settlements and other types of infrastructure commonly occur in valleys and lower elevations where permafrost is not present. This may give an impression that central infrastructure is missing. Moreover, owing to its public-participatory nature, the OpenStreetMap data can lack objects but we are not aware of any more comprehensive global databases that include all types of populated settlements.

Reviewer #3: Page 24 Eq. 1 and 2: I am not familiar with standard notations for this type of statistical modelling, but it seems inappropriate to simply add the parameters (TDD, FDD, SOC, etc.). These have been defined in the preceding text as physical and environmental parameters, quantifiable and with specified units (e.g., deg*days for TDD and FDD, g/kg for SOC etc.). Could the equations be adapted to make it clear that the statistical model used is a function of these predictor variables? (and would the function be different for the four different statistical models used?)

R: Equations (1) and (2) were removed during the revision (please see the response to Reviewer #1).

Reviewer #3: Page 39 Fig 3a: Only the GAM plot has a lower whisker - why?

R: This figure was removed during the revision (please see above).

Reviewer #3: Page 39 Fig 3c: This plot seems almost identical to Manuscript (1) figure 2b - but statistics are somewhat different, why?

R: This figure was removed during the revision (please see above).

Reviewer #3: Page 42 Fig 6: See comments above for Page 17 Fig 4.

R: We improved the figure according to the comments (see above).

Reviewer #4 (Remarks to the Author):

Reviewer #4: Thank you for giving me the opportunity to review this interesting manuscript. Below, I provide a number of misc. comments for your consideration:

*Lines 53-54: Please provide a reference citing the relatively short lifespan of Arctic infrastructure

R: A reference was added: (13) ACIA. Impacts of a Warming Arctic: Arctic Climate Impact Assessment (Cambridge Univ. Press, 2004).

Reviewer #4: *Lines 83-85: At least one of these pipelines was designed with the use of thermosyphons--to mitigate some or all of the impact of thawing permafrost. Is this technology or other technologies accounted for in your analysis? It appears that a major shortcoming may be the fact that they assume that communities do not adapt to some (or all) of these changes.

R: We shortly covered this topic in the introduction "... the assessment should be complemented by local-scale process modelling, taking into account site-specific engineering, design, and construction practices³") and discussion "...Although engineering solutions can address both human-induced and naturally caused problems, their economic cost may be prohibitive at regional scales²⁹...."). However, following your comment, we made small additions to the introduction (lines 71–72, addition underlined: "...taking into account site-specific engineering, design, and construction practices (e.g. adaptation strategies)^{3,10,12–14}") and discussion (lines 150–153: "...Although engineering solutions (e.g. adaptation strategies and structures such as insulation and thermosyphons that were not considered in this pan-Arctic study) can address to some extent both human-induced and naturally caused problems...").

Please note that we added "to some extent" above because these engineering solutions are still limited in terms of withstanding climate warming. They can keep the structure stable only to some certain limit of ground warming and thawing everywhere around the structure. Also, none of these engineering solutions usually can prevent development of hazardous slope processes (caused by warming) in the vicinity of the engineering structures being protected.

Reviewer #4: *Lines 133-135: Damage to critical energy delivery and industrial infrastructure can also impact broader economic activity and national security.

R: Following the comment, we added a sentence on this important topic (lines 139–140): "Moreover, damage to critical energy delivery and industrial infrastructure can affect general economic activity and national security¹²⁻¹⁴."

Reviewer #4: *Reference #13: I do not think it is appropriate to include a reference to a newspaper article--cite the original study if it has been independently peer-reviewed.

R: The reference was removed. An additional reference was not introduced to reduce the amount of references. In general, owing to the changes elsewhere a total of 21 references were removed to follow the guideline (max 70 references should be included).

Reviewer #4: *Little or no discussion was provided on how accurate the infrastructure count was across the Arctic (compared to other studies that have considered infrastructure at risk to permafrost degradation). For example, a study by Larsen et al. (2008)--Global Environmental Change--indicated that there was considerable uncertainty about the count and location of infrastructure

located on permafrost across the U.S. state of Alaska. That study's infrastructure database was built using a bottoms-up data collection approach.

R: In general, considering spatial accuracy, the quality of the main source for infrastructure data (OpenStreetMap, www.openstreetmap.org) is seen to be comparable to commercial or locally produced data products (e.g. Haklay et al., 2010; Zhang & Malczewski, 2017). To our knowledge, there exist no assessments for pan-Arctic area nor circumpolar permafrost domain, and such an extensive assessment is beyond the scope of this study. Considering the scale of analysis (ca. 1 km) and extent of the study area (the whole Northern Hemisphere permafrost domain) but also data quality assessments conducted in other regions, we consider the used infrastructure data to be the best publicly available and suitable for the purpose. Owing to sparse population and remoteness of permafrost areas, there could be more inaccuracies in the Arctic compared to more densely populated regions. To address uncertainty of infrastructure data sets (especially OSM because it was the main source), we added a short discussion to Methods (lines 335–341):

“Geospatial data quality encompasses many aspects, e.g. location accuracy, completeness of elements or their attributes, which in the case of OSM have been extensively studied predominantly in highly developed areas^{43,44}, whereas in remote regions these evaluations are scarce. Recently, Ref. (50) estimated that in 2015 global OSM road network was ~83% complete albeit between-country differences were identified. Here, we included only five top-tier road types, as opposed to smaller roads included in their analysis⁵⁰, which was assumed to reduce the risk of data quality discrepancies between regions⁴⁶.”

(43) Haklay, M., Basiouka, S., Antoniou, V. & Ather, A. How many volunteers does it take to map an area well? The validity of Linus' Law to volunteered geographic information. *Cartogr. J.* **47**, 315–322 (2010).

(44) Zhang, H. & Malczewski, J. in *Volunteered Geographic Information and the Future of Geospatial Data* (eds. Campelo, C. E. C., Bertolotto, M. & Corcoran, P.) (IGI Global, 2017).

(46) OECD/ITF. *ITF Transport Outlook 2017* <http://dx.doi.org/10.1787/9789282108000-en> (OECD Publishing Paris, 2017)

(50) Barrington-Leigh, C. & Millard-Ball, A. The world's user-generated road map is more than 80% complete. *PloS One* **12**, e0180698 (2017).

Reviewer #4: *My sense is that the uncertainty in the location, type, value and amount of infrastructure is larger than the uncertainty in the modeling of the permafrost under different climate scenarios. This statement is consistent with other studies which point to the uncertainty in socio-economic variables exceeding the uncertainty in the geophysical variables.

R: Elements of infrastructure are physical structures and can be observed directly. Considering this infrastructure data should include less uncertainty in the location, type, and quantity than most of the socio-economic variables that cannot be measured or determined directly. To minimize heterogeneity of the infrastructure data, we used broadly utilized publicly available global products with comparable high spatial accuracy. Please see also the response to the previous remark.

Reviewer #4: *This analysis did not appear to account for population changes between now and the middle-of-the-century. We know that some places in the rural Arctic are reporting decreasing

populations over time--despite overall population growth globally projected into the middle-of-the-century.

R: It is true that population of several Arctic settlements has changed, commonly decreased. However, we are not aware of any suitable population projection that would match with the extent of our study area, resolution and periods analysed. Thus, we consider that using current population counts is the safest way to address this issue associated with unpredictable aspects of near-future socio-economic development. We added a short explanation on this issue in the Method section (lines 327–331):

“As we are not aware of any population projection that would match with the extent of our study area, resolution and periods analysed, we consider that using current population counts is the safest way to address human exposure to future hazards, even though changes in population, as well as in infrastructure, are probable but subject to (unpredictable) near-future socio-economic development.”

Reviewer #4: *Finally, the above implies that the authors performed a rigorous analysis on the future state of the climate/permafrost, but did not consider future population or changes in infrastructure over the next 30-40 years. These changes may be considerable--yet they do not appear to have been accounted for in this analysis.

R: Please see the response above. This issue was also considered in the discussion section (lines: 156–160) as an option in future studies, if adequate datasets would be available (addition underlined): “The major advantage of the approach presented here is that hazard quantification can be conducted with any available infrastructure or population dataset (also using planned infrastructure and future population if suitable high-quality datasets and projections are available) and for any policy-relevant global warming scenario.”

Reviewers' comments:

Reviewer #1 (Remarks to the Author):

I recognize that this is a resubmission. The authors have revised some of comments from me. But, there are a question that I always think it decrease the innovativeness and contribution of this manuscript. It is that the panel (d) in the figure 2 of this manuscript is the same as the panel (a) in figure 2 of the published paper (doi: 10.1029/2018GL078007). Also, the extended data figure 2 is the same as the Figure 3 of the related manuscript file (Circumpolar permafrost maps and geohazard indices for near-future infrastructure risk assessments). I do not know whether the question can be permitted by the journal of Nature Communications. I give the judge whether this manuscript can be accepted to the editor of this Journal.

Reviewer #2 (Remarks to the Author):

Replies to reviewers' comments on manuscript NCOMMS-18-06236

"Degrading permafrost puts Arctic infrastructure at risk by mid-century" answered the questions well. I recommend the revised manuscript to be polished.

Reviewer #3 (Remarks to the Author):

I have reviewed the revised version of the paper titled "Degrading permafrost puts Arctic infrastructure at risk by mid-century".

I find that the revision has resulted in a much improved paper with a clear focus on the infrastructure hazard evaluation, and the issues of overlap with other papers have been resolved. All my previous comments have been satisfactorily implemented or disputed with acceptable arguments, and figures have been improved to be more readable and more readily understandable.

In the revised version, it is made clear that the condition for near surface permafrost thaw is that MAGT at depth of zero annual amplitude (d_{zaa}) increases to $>0C$. This threshold is used in a separate zonation and evaluation (section 1, P5 L74) describing the amount of infrastructure in areas where near surface permafrost will disappear within the timeframe evaluated, and may be a reasonable threshold in this context.

However, substantial ground ice melt will occur before MAGT at d_{zaa} changes to $>0C$. I therefore consider the choice of threshold parameter very (extremely) conservative in an engineering context and it is probably not a good practical indicator for thaw settlements affecting infrastructure. Most infrastructure would experience severe thaw settlements and failure before this condition is met and its use in the hazard classification is therefore questionable. I would like the authors to briefly mention this problem in the methods section.

P23 L391-395

Please specify again here that MAGT is taken at the depth of zero annual amplitude. The authors should probably also state how d_{zaa} is defined (variation less than $0.1C?$).

Based on this, and my previous appraisal of the content and scope of the paper (repeated below) I recommend the paper for publication

The authors present a very thorough and well founded quantitative analysis of the impact of climate change on infrastructure hazards in Arctic permafrost areas.

To my knowledge, this study is the first to quantify the different types of infrastructure at risk on a circum-Arctic scale under different climate change scenarios. By quantifying infrastructure hazards at this scale, the study provides decision makers at all levels with a valuable tool in the evaluation and prioritizing of adaptation and mitigation measures.

The study is very thorough in its design at all levels: The forecasts of permafrost change is an ensemble average of four independent statistical methods applied to the prediction of mean annual ground temperature (MAGT) and active layer thickness (ALT). Similarly, the hazard mapping is based on three different hazard evaluation models, the results of which are consolidated through a majority-vote approach in a final consensus index to be used in the quantification of infrastructure at risk.

The data basis for both statistical modelling and hazard mapping is harvested from well-established, international data networks, such as the GTN-P, combined with local and national sources, and the process is clearly described and data sources and types listed in supplementary tables.

It is my evaluation that both methodology and the data basis are valid and of high quality, and that the study is highly relevant and recommendable for publication(s).

Reviewer #4 (Remarks to the Author):

I am pleased to provide a review of "Degrading permafrost puts Arctic infrastructure at risk by mid-century" by Hjort et al.

In general, this article is very well-written—it is easy to read and the graphics are quite interesting. I liked that the authors conducted a pan-Arctic analysis, but then focused on hydrocarbon extraction fields in Russia in a "special investigation". An important omission, which the authors acknowledge, is the fact that no adaptation strategies were considered. The results should be presented with and without an adaptation scenario. It is not realistic, even at regional scales, to assume that communities will not attempt to adapt in some way to the observed changes. I also question the validity of using one, selected reference to indicate that the "economic costs [of adaptation] may be prohibitive at regional scales". Furthermore, the authors indicate that potential harm to pipelines and industrial facilities can be larger than estimated, but then they fail to acknowledge that a number of these facilities, including the Trans-Alaska pipeline have thermosyphons installed along sections that are built on top of (or near) permafrost. The companies that maintain these pipelines and industrial facilities typically have the financial resources to address environmental hazards, including permafrost thaw, to some degree. I do not believe this study should be published until the authors have considered some sort of adaptation scenario. Adaptation scenario analyses for the Arctic have been demonstrated in Larsen et al. (2008), Chinowsky et al. (2009), and Melvin et al. (2016).

A key uncertainty, which the authors acknowledge, is the count and location of infrastructure compiled via the OSM and WikiProject sources. It would be useful to see a little bit more of a rigorous discussion about these sources, their accuracy for a few case study communities, etc.

The Hjort et al. manuscript could be significantly improved by (1) showing a distinct lineage in the past literature on this subject (Arctic infrastructure at risk), (2) highlight how they have improved upon the existing work of earlier researchers, and (3) conclude with additional areas of research

needed (see below).

The following are some more specific comments on the manuscript:

*Page 7, Line 128: The authors use terms to communicate a degree of uncertainty (“moderate”), but this statement appears to be the authors’ opinion and not a precise statement on the relative accuracy of the projections of ground temperature and annual thaw depth versus other sources of uncertainty. Moderate uncertainty compared to what?

*Page 3, Line 40: Would be helpful to identify right up-front what the “key pieces of knowledge [that] are still missing” and then conclude with some explicit statements about what additional research is needed. For example, Larsen et al. (2008) manuscript in *Global Environmental Change*—not cited in this study—estimated costs to Alaska infrastructure from projected climate change. They suggested three topics for future study. The Melvin et al. (2016) study, which is cited here, then took the recommendations of Larsen et al. (2008) and improved the cost estimation for Alaska infrastructure.

- Chinowsky, P., K. Strzepek, P. Larsen, and A. Opdahl. 2009. Adaptive climate response cost models for infrastructure. *Journal of Infrastructure Systems* 16(3), 173-225.
- Larsen, P., S. Goldsmith, O. Smith, M.L. Wilson, K. Strzepek, P. Chinowsky and B. Saylor. 2008. Estimating future costs for Alaska public infrastructure at risk from climate change, *Global Environmental Change* 18 (2008), 442–457.

Replies to reviewers' comments on manuscript NCOMMS-18-06236A "Degrading permafrost puts Arctic infrastructure at risk by mid-century"

Our response (**R**) to the **Reviewer** comments appear below. Line numbering in the responses refer to the PDF version of the revised manuscript (not the Word document with Track Changes).

Reviewer #1 (Remarks to the Author):

Reviewer #1: I recognize that this is a resubmission. The authors have revised some of comments from me. But, there are a question that I always think it decrease the innovativeness and contribution of this manuscript. It is that the panel (d) in the figure 2 of this manuscript is the same as the panel (a) in figure 2 of the published paper (doi: 10.1029/2018GL078007).

R: Thank you for your observation. Although there is some overlap between the indicated panels, they are not the same. Figure 2d of this manuscript shows the hindcast results for 1970-1984 and Figure 2a in Aalto et al. (2018) shows the hindcast results also for 1985-1999 and using partly different evaluation measure. However, to remove the confusion related to these two subfigures we decided to delete the scatter plots from the Figure 2 of this manuscript.

Reviewer #1: Also, the extended data figure 2 is the same as the Figure 3 of the related manuscript file (Circumpolar permafrost maps and geohazard indices for near-future infrastructure risk assessments). I do not know whether the question can be permitted by the journal of Nature Communications. I give the judge whether this manuscript can be accepted to the editor of this Journal.

R: Actually, there are certain differences (e.g., the extent of the circumpolar area is different), although the presented geohazard indices are the same. To remove the overlap, Figure 3 of the related manuscript will be deleted and replaced by a new figure. Thus, we kept the current Extended data Figure 2 as it is.

Reviewer #2 (Remarks to the Author):

Reviewer #2: Replies to reviewers' comments on manuscript NCOMMS-18-06236 "Degrading permafrost puts Arctic infrastructure at risk by mid-century" answered the questions well. I recommend the revised manuscript to be pulished.

R: We are happy to hear that we managed to address all the raised issues well. Thank you for your positive response and recommendation for publishing our work in Nature Communications.

Reviewer #3 (Remarks to the Author):

Reviewer #3: I have reviewed the revised version of the paper titled "Degrading permafrost puts Arctic infrastructure at risk by mid-century".

I find that the revision has resulted in a much improved paper with a clear focus on the infrastructure hazard evaluation, and the issues of overlap with other papers have been resolved. All my previous comments have been satisfactorily implemented or disputed with acceptable arguments, and figures have been improved to be more readable and more readily understandable.

R: Thank you for your effort to improve the manuscript and positive response.

Reviewer #3: In the revised version, it is made clear that the condition for near surface permafrost thaw is that MAGT at depth of zero annual amplitude (d_{zaa}) increases to $>0^{\circ}\text{C}$. This threshold is used in a separate zonation and evaluation (section 1, P5 L74) describing the amount of infrastructure in areas where near surface permafrost will disappear within the timeframe evaluated, and may be a reasonable threshold in this context.

However, substantial ground ice melt will occur before MAGT at d_{zaa} changes to $>0^{\circ}\text{C}$. I therefore consider the choice of threshold parameter very (extremely) conservative in an engineering context and it is probably not a good practical indicator for thaw settlements affecting infrastructure. Most infrastructure would experience severe thaw settlements and failure before this condition is met and its use in the hazard classification is therefore questionable. I would like the authors to briefly mention this problem in the methods section.

R: We addressed the issue by adding two new sentences in the main text of the manuscript (lines 63–66): "...In an engineering context, the selected threshold is conservative because infrastructure (e.g., buildings) could experience thaw settlements and failure before the thaw of near-surface permafrost. However, a conservative threshold is justified considering the use of statistically-based methodology in modelling of the ground thermal regime (Methods)."

Reviewer #3: P23 L391-395

Please specify again here that MAGT is taken at the depth of zero annual amplitude.

R: Specified as recommended.

Reviewer #3: The authors should probably also state how d_{zaa} is defined (variation less than 0.1°C ?).

R: A definition for ZAA is given in the Method section (lines 285–286): "MAGT observations at or near (the closest to) the depth of zero annual amplitude (ZAA, annual temperature variation $< 0.1^{\circ}\text{C}$)³ were utilized"

Reviewer #3: Based on this, and my previous appraisal of the content and scope of the paper (repeated below) I recommend the paper for publication

''''''

The authors present a very thorough and well founded quantitative analysis of the impact of climate change on infrastructure hazards in Arctic permafrost areas.

To my knowledge, this study is the first to quantify the different types of infrastructure at risk on a circum-Arctic scale under different climate change scenarios. By quantifying infrastructure hazards at this scale, the study provides decision makers at all levels with a valuable tool in the evaluation and prioritizing of adaptation and mitigation measures.

The study is very thorough in its design at all levels: The forecasts of permafrost change is an ensemble average of four independent statistical methods applied to the prediction of mean annual ground temperature (MAGT) and active layer thickness (ALT). Similarly, the hazard mapping is based on three different hazard evaluation models, the results of which are consolidated through a majority-vote approach in a final consensus index to be used in the quantification of infrastructure at risk.

The data basis for both statistical modelling and hazard mapping is harvested from well-established, international data networks, such as the GTN-P, combined with local and national sources, and the process is clearly described and data sources and types listed in supplementary tables.

It is my evaluation that both methodology and the data basis are valid and of high quality, and that the study is highly relevant and recommendable for publication(s).

''''''

R: We are pleased to get these positive comments. Thank you very much recommending our work for publication.

Reviewer #4 (Remarks to the Author):

Reviewer #4: I am pleased to provide a review of “Degrading permafrost puts Arctic infrastructure at risk by mid-century” by Hjort et al.

In general, this article is very well-written—it is easy to read and the graphics are quite interesting. I liked that the authors conducted a pan-Arctic analysis, but then focused on hydrocarbon extraction fields in Russia in a “special investigation”.

R: Thank you for your substantial effort to improve the manuscript and constructive comments.

Reviewer #4: An important omission, which the authors acknowledge, is the fact that no adaptation strategies were considered. The results should be presented with and without an adaptation scenario. It is not realistic, even at regional scales, to assume that communities will not attempt to adapt in some way to the observed changes.

R: It is true that the adaptation issue deserves more attention, but from a different angle. Still, thank you for pointing this out. First, we would politely like to reiterate the aims of our work: (1) to identify (i.e., **map**) at unprecedentedly high spatial resolution **infrastructure hazard areas** in the Northern Hemisphere’s permafrost regions; and (2) to quantify the amount and proportion of fundamental engineering structures **existing** in areas where ground subsidence and loss of structural bearing capacity could damage infrastructure (i.e., are at risk) by 2050. These were clearly presented in the abstract but we realized that the aims could also have been highlighted better in the introductory section (lines 44–48): “The aim of this study was to (i) map infrastructure hazard areas in the Northern Hemisphere’s permafrost regions at unprecedentedly high (~1 km) spatial resolution under projected climatic changes and (ii) quantify the amount and proportion of engineering structures in areas where ground subsidence and loss of structural bearing capacity could damage infrastructure by 2050.”

Second, it is true that communities can and should try to adapt to challenging building conditions. However, our results (e.g., how many settlements are located in high hazard zone) are not dependent on how communities can (or cannot) adapt to the adverse changes because the same amount of infrastructure will exist in a certain hazard zone, with or without adaptation. The research aims were reflected in the presentation of conclusions: “A total of 69% of the pan-Arctic residential, transportation, and industrial **infrastructure is located in areas** with high potential for near-surface permafrost thaw by 2050. Consideration of ground properties in addition to permafrost thaw showed that 33% of **infrastructure is located in areas** where ground subsidence and loss of structural bearing capacity could severely damage the integrity of infrastructure.”

Third, our study is particularly focused on the identification of permafrost thaw and high hazard potential in areas where consideration of adaptation is especially relevant in the near future. This was presented in the previous revision, but we have tried to communicate this better in the second revision in the end of the introduction (“Our study reveals the magnitude of the threat to engineering structures from climate change at the pan-Arctic scale, and show where detailed infrastructure risk assessment should be conducted in the near future.”) and discussion (“Our study focused on a pan-Arctic assessment with the goal of showing where regional and local-scale risk assessments, taking into account site-specific engineering, design, and construction practices (e.g., adaptation strategies)^{2,9,11,13,14} should be conducted in the near future.”).

This study is the first step of a strategy to better consider the adaptation issues highlighted in the recent AMAP report [AMAP. Snow, Water, Ice and Permafrost in the Arctic (SWIPA). (Arctic Monitoring and Assessment Programme (AMAP), Oslo, Norway, 2017)]. This was noted in the original manuscript (lines 132–134; “This result is congruent with projected changes in the Arctic¹, and emphasizes the need for adaptation-based policies at community and regional levels in the near future...” and in the revised discussion (lines 188–191; “...locally and regionally applied mitigation strategies for existing infrastructure and future development projects are paramount for sustainable development in the Arctic¹. Our study can be considered to be a step forward toward these goals.”).

Reviewer #4: I also question the validity of using one, selected reference to indicate that the “economic costs [of adaptation] may be prohibitive at regional scales”.

R: We added two other references (AMAP 2011; Melvin et al. 2016) to support the statement. For example, Melvin et al. (2016) state that: “No adaptation measures for permafrost thaw were identified that were less expensive than complete infrastructure replacement (p. E125)” and “Methods for adapting infrastructure to near-surface permafrost thaw are limited and costly (p. E125)”. Please also note the expression “may be”.

Reviewer #4: Furthermore, the authors indicate that potential harm to pipelines and industrial facilities can be larger than estimated, but then they fail to acknowledge that a number of these facilities, including the Trans-Alaska pipeline have thermosyphons installed along sections that are built on top of (or near) permafrost. The companies that maintain these pipelines and industrial facilities typically have the financial resources to address environmental hazards, including permafrost thaw, to some degree.

R: We quantified the proportion of industrial infrastructure in different hazard zones. As presented above, our result is not dependent on adaptation measures because the same amount of infrastructure will still occur in, for example, high hazard zone with adaptation (e.g., pipelines will probably remain intact) or without adaptation (e.g., pipelines may be damaged) to address permafrost thaw-related ground subsidence. Based on our results (i.e., how much industrial infrastructure is present in a high hazard zone in an ‘average’ situation compared to the worst and best case), we consider our statement to be fair. However, we removed the word “substantially” to soften the expression. Please also note that we used the expression “potential harm” in this example.

Reviewer #4: I do not believe this study should be published until the authors have considered some sort of adaptation scenario. Adaptation scenario analyses for the Arctic have been demonstrated in Larsen et al. (2008), Chinowsky et al. (2009), and Melvin et al. (2016).

R: We carefully read these (and some related papers) again. [The Chinowsky et al. paper we found was from 2010 but most likely we used the same paper as you suggested.] The economic assessments presented in the listed papers are highly interesting and comprehensive but they are beyond the scope of our work. Please see the responses above. Moreover, consideration of a scientifically valid adaptation scenario would require a completely new study or rather a series of studies at our pan-Arctic scale. For example, building principles vary from country to country (or from region to region) and this, among many other site- and region-specific facts (e.g., Alaska is

intensively mapped and most of the other Arctic areas are not; there are no comparable background data available for many other regions), complicate the analysis substantially.

Reviewer #4: A key uncertainty, which the authors acknowledge, is the count and location of infrastructure compiled via the OSM and WikiProject sources. It would be useful to see a little bit more of a rigorous discussion about these sources, their accuracy for a few case study communities, etc.

R: Owing to the lack of homogeneous (i.e., uniform quality across nations and regions) circumpolar infrastructure data for comparison, we cannot comprehensively assess the potential deficiencies and inaccuracies of the data sets that were used. However, we made new analyses and added new text (underlined) to better consider uncertainty issues (line 358–377):

“Geospatial data quality encompasses many aspects, e.g. location accuracy, completeness of elements or their attributes, which in the case of OSM have been extensively studied predominantly in highly developed areas^{45,46}, whereas in remote regions these evaluations are scarce. The circumpolar applicability of OSM road and railway data, has been demonstrated by their previous use in the production of global 100-m resolution grids for socioeconomic/population (WorldPop Project⁵²) and global travel time to cities mapping at 1 km resolution⁵³. Ref. (54) estimated that in 2015 global OSM road network was ~83% complete albeit between-country differences were identified. Here, we included only five top-tier road types, as opposed to smaller roads included in their analysis⁵⁴, which was assumed to reduce the risk of data quality discrepancies between regions⁴⁸. Apart from roads, very few global-scale evaluations of the OSM data have been performed. Moreover, no systematic framework to evaluate OSM data yet exists⁵⁵.

According to our calculations, the total length of WikiProject pipelines in Russia (baseline permafrost conditions) is ~5% greater than those in the federal Rosnedra database (gis.sobr.geosys.ru). This is attributed to higher spatial resolution and a more detailed presentation of pipeline networks within communities and oil/gas fields. Compared to the documented lengths of a few central pipeline systems (including non-permafrost areas), the data encompass 99.8% of TAPS (1,285/1,288 km, akpipelinesafety.org), 98.9 % of ESPO (4,702/4,756 km, energybase.ru), 93.1 % of Urengoy–Pomary–Uzhgorod pipeline (4,142/4,451 km) and 76.2 % of Bovanenkovo–Ukhta–Torzhok (2,009/2,637 km), suggesting that they are geospatially mostly complete and accurate.

The analyses involving buildings presented here are preliminary, as the number of OSM buildings across our modelled permafrost domain was obviously much less than the actual number. Moreover, region-specific differences exist. A simple people-per-building -ratio (regional population divided by number of buildings) was calculated to provide a rough estimate of the validity of building counts in the geographical regions under consideration. Eurasia and North America had reasonable ratios, 23 and 12, respectively, while for central Asian mountains a ratio of nearly 700 indicated that a large number of buildings could be missing. Urban settlements, which contain the majority of buildings, had good coverage, while in some of the smaller populated places infrastructure may not have been mapped. In the context of this study, which includes all settlements ranging from isolated dwellings to cities, it is important to take into account the maximum extent of human activities. This was achieved with the OSM ‘places’ map feature, which includes ca. 10 times more populated settlements than analogous open datasets (e.g., the Global Rural Urban Mapping Project, Naturalearthdata.com – Populated places) across the modelled permafrost region.”

Reviewer #4: The Hjort et al. manuscript could be significantly improved by (1) showing a distinct lineage in the past literature on this subject (Arctic infrastructure at risk), (2) highlight how they have improved upon the existing work of earlier researchers, and (3) conclude with additional areas of research needed (see below).

R: Following these good suggestions, we (1) modified the introduction and added key references on Arctic infrastructure issues (lines 37–42; please see response below), (2) highlighted what new our work provides compared to previous ones (discussion section; lines 167–170) and (3) concluded what could be important next steps in pan-Arctic infrastructure hazard and risk assessment in future (discussion section; lines 176–183). Our additions were intentionally relatively short to keep the text concise and reader friendly for a wide readership (new text underlined):

“...Consequently, detailed hazard maps and geospatial data-based computations, such as those presented here, are of importance to enable planners and policy-makers to identify both high- and low-hazard areas when planning future infrastructure at urban and settlement scales^{2,9,17,33}. Our analyses were conducted at a higher spatial resolution than previous studies^{17,19–21}, and the results presented here are based on a consensus of three different indices (see Methods). Moreover, we were able to quantify and show the magnitude of infrastructure at risk across the circumpolar permafrost domain. The major advantage of the approach presented here is that hazard quantification can be conducted with any available infrastructure or population dataset (also using planned infrastructure and future population if suitable high-quality datasets and projections are available) and for any policy-relevant global warming scenario.

Our study focused on a pan-Arctic assessment with the goal of showing where regional and local-scale risk assessments, taking into account site-specific engineering, design, and construction practices (e.g., adaptation strategies)^{2,9,11,13,14} should be conducted in the near future. The forthcoming infrastructure risk assessments would significantly benefit from applicable process-based transient models of ground thermal regime and high-resolution climate and ground-ice data. With the help of improved permafrost projections, hazard maps and verified infrastructure data it would be feasible to quantify the economic impacts of climate change on infrastructure at the pan-Arctic scale (e.g., following Ref. (9)).

To successfully manage climate change impacts in sensitive permafrost environments, a better understanding is needed about which elements of the infrastructure are likely to be affected by climate change, where they are located, and how to implement adaptive management in the most effective way, considering the changing environmental conditions. Such locally and regionally applied mitigation strategies for existing infrastructure and future development projects are paramount for sustainable development in the Arctic¹. Our study can be considered to be a step forward toward these goals.”

Reviewer #4: The following are some more specific comments on the manuscript:

*Page 7, Line 128: The authors use terms to communicate a degree of uncertainty (“moderate”), but this statement appears to be the authors’ opinion and not a precise statement on the relative accuracy of the projections of ground temperature and annual thaw depth versus other sources of uncertainty. Moderate uncertainty compared to what?

R: The text was revised to remove the unspecified expression “a moderate amount of uncertainty”.

Reviewer #4: *Page 3, Line 40: Would be helpful to identify right up-front what the “key pieces of knowledge [that] are still missing” and then conclude with some explicit statements about what additional research is needed. For example, Larsen et al. (2008) manuscript in *Global Environmental Change*—not cited in this study—estimated costs to Alaska infrastructure from projected climate change. They suggested three topics for future study. The Melvin et al. (2016) study, which is cited here, then took the recommendations of Larsen et al. (2008) and improved the cost estimation for Alaska infrastructure.

- Chinowsky, P., K. Strzepek, P. Larsen, and A. Opdahl. 2009. Adaptive climate response cost models for infrastructure. *Journal of Infrastructure Systems* 16(3), 173-225.

- Larsen, P., S. Goldsmith, O. Smith, M.L. Wilson, K. Strzepek, P. Chinowsky and B. Saylor. 2008. Estimating future costs for Alaska public infrastructure at risk from climate change, *Global Environmental Change* 18 (2008), 442–457.

R: We improved the text to clarify the gap in knowledge (new text underlined):

“Arctic natural and anthropogenic systems are undergoing unprecedented changes¹, with permafrost thaw as one of the most striking impacts in the terrestrial cryosphere^{2–4}. In addition to the potential adverse effects on global climate⁵, ecosystems⁶, and human health⁷, warming and thaw of near-surface permafrost may impair critical infrastructure^{8,9} (Fig. 1). This could pose a serious threat to the utilization of natural resources¹⁰, and to the sustainable development of Arctic communities^{9,11,12}. Extensive summaries of damage to infrastructure along with adaptation and mitigation strategies are available^{11,13–18}. Benchmark reports^{1,13,14} call for pan-Arctic geohazard explorations and infrastructure risk assessments, but only regional studies^{17,19–21} have been conducted since Ref. (8). There is an urgent need for pan-Arctic geohazard mapping at high spatial resolution and an assessment of how changes in circumpolar permafrost conditions could affect infrastructure^{1,14}. Owing to the increasing economic and environmental relevance of the Arctic^{1,5,10}, it is of a vital importance to gain detailed knowledge about risk exposure in areas of current and future infrastructure^{8–14,18}. The aim of this study was to (i) map infrastructure hazard areas in the Northern Hemisphere’s permafrost regions at unprecedentedly high (~1 km) spatial resolution under projected climatic changes and (ii) quantify the amount and proportion of engineering structures in areas where ground subsidence and loss of structural bearing capacity could damage infrastructure by 2050.”

As you noted, Melvin et al. (2016) was cited in our manuscript but Larsen et al. (2008) not. The latter paper has been an important background paper during the planning of our work. For some reason a reference to this paper was missing, but is now added.

REVIEWERS' COMMENTS:

Reviewer #4 (Remarks to the Author):

The authors have addressed all of my suggestions to my satisfaction. Thank you for taking the time to revise this important and timely paper.

Reply to reviewer comment on manuscript NCOMMS-18-06236B
"Degrading permafrost puts Arctic infrastructure at risk by mid-century"

Our response (**R**) to the **Reviewer** comment appear below.

Reviewer #4 (Remarks to the Author):

Reviewer #4: The authors have addressed all of my suggestions to my satisfaction. Thank you for taking the time to revise this important and timely paper.

R: We are happy to hear that we managed to address all the raised issues. Thank you for your substantial effort to improve the manuscript.